# Evolution of events before and after the 17 June 2017 rock avalanche at Karrat Fjord, West Greenland – a multidisciplinary approach to detect and locate unstable rock slopes in a remote Arctic area

Kristian Svennevig[1], Trine Dahl-Jensen[1], Marie Keiding[1], John Peter Merryman Boncori[2], Tine B. Larsen[1], Sara Salehi[1], Anne Munck Solgaard[1] and Peter H. Voss[1]

[1]Geological Survey of Denmark and Greenland (GEUS), Copenhagen, 1350, Denmark.

[2] DTU Space, National Space Institute, Technical University of Denmark, Lyngby, 2800, Denmark.

*Correspondence to*: Kristian Svennevig (ksv@geus.dk)

**Abstract**

The 17 June 2017 rock avalanche in the Karrat Fjord, West Greenland caused a tsunami that flooded the nearby village of Nuugaatsiaq and killed four persons. The disaster was entirely unexpected since no previous records of large rock slope failures were known in the region, and it highlighted the need for a better knowledge of potentially hazardous rock slopes in remote Arctic regions.

The aim of the paper is to explore our ability to detect and locate unstable rock slopes in remote Arctic regions with difficult access. We test this by examining the case of the 17 June 2017 Karrat rock avalanche. The workflow we apply is based on a multidisciplinary analysis of freely available data comprising seismological records, Sentinel-1 space borne Synthetic Aperture Radar (SAR) data and Landsat and Sentinel-2 optical satellite imagery, ground truthed with limited fieldwork. Using this workflow enables us to reconstruct a timeline of rock slope failures on the coastal slope here collectively termed the Karrat Landslide Complex.

Our analyses show that at least three recent rock avalanches occurred in the Karrat Landslide Complex: Karrat 2009, Karrat 2016 and Karrat 2017. The latter is the source of the abovementioned tsunami, whereas the first two are described here in detail for the first time. All three are interpreted to have initiated as dipslope failures. In addition to the recent rock avalanches, older rock avalanche deposits are observed, demonstrating older (Holocene) periods of activity. Furthermore, three larger unstable rock slopes that may pose a future hazard are described. A number of non-tectonic seismic events confined to the area are interpreted to record rock slope failures. The structural setting of the Karrat Landslide Complex, namely dipslope, is probably the main conditioning factor for the past and present activity and based on the temporal

distribution of events in the area, we speculate that the possible trigger for rock slope failures is permafrost degradation caused by climate warming.

The results of the present work highlight the benefits of a multidisciplinary approach based on freely available data to study unstable rock slopes in remote Arctic areas under difficult logistical field conditions and demonstrate the importance

of identifying minor precursor events to identify areas of future hazard.

## Introduction

On 17 June 2017 the village of Nuugaatsiaq in West Greenland was hit by a tsunami generated by a 35-58 million $m^3$ rock avalanche on the south facing slope of the Ummiammakku mountain in Karrat Fjord, located 32 km to the east of

the village (Bessette-Kirton et al., 2017; Clinton et al., 2017; Gauthier et al., 2018; Paris et al., 2019). A large part of the village was destroyed, and four people lost their lives. The tsunami was also observed in other settlements more than 100 km away. Following this, the Greenlandic authorities evacuated 170 residents from Nuugaatsiaq and the neighbouring settlement of Illorsuit due to the threat of further rock slope failures in the area and the villages are still under evacuation orders at the time of this writing due to fear of additional induced tsunamis (Fig. 1).

Rock slope failures were not known from the Karrat Fjord prior to the 2017 rock avalanche. This highlighted the necessity to screen the inhabited parts of Greenland for unstable rock slopes and to document previous large rock avalanches to assess the threat from future tsunamigenic events. Two tsunami-generating rock avalanches in 1952 and 2000 are described from Vaigat 150 km to the south of Karrat (Pedersen et al., 2002; Dahl-Jensen et al., 2004). Svennevig (2019) described morphological evidence of several Holocene rock slope failures in the region but noted that the majority of

these were located in the area of the Cretaceous-Paleogene Nuussuaq Basin, where the 1952 and 2000 rock avalanches also occurred. The geological province where the 2017 rock avalanche occurred was found to have had relatively few rock slope failures (Fig. 1A).

Fieldwork and in situ measurements are difficult and time consuming in a vast and remote Arctic environment like Greenland where infrastructure is minimal and expensive. Thus, investigations of unstable slopes over large parts of

Greenland must primarily rely on remote sensing techniques. Following the launch of Landsat-8 and Sentinel-1 and 2 satellites, optical and ground motion data over Greenland are both free and frequent and at sufficient resolution, providing a means of observing unstable slopes at relatively low cost. Svennevig et al. (2019) preliminarily described a multi-disciplinary approach combining satellite data and seismic observations to remotely study activity on an unstable slope. They found that by combining these methods, it was possible to reliably detect timing (seismic observations), location,

extent and deformational rates (optical images, ground motion observations from DInSAR (Differential Interferometric Synthetic Aperture Radar)) of rock slope failures.

Our aims with this study are twofold: 1) to understand the processes that led to the disastrous Karrat 2017 rock avalanche and the continued threat from the area, 2) and to explore our ability to detect and locate rock slope failures and ultimately to assess the associated hazard in an unhospitable climate with very difficult access. Following the multi-disciplinary

approach preliminarily described by Svennevig et al. (2019), it is possible to resolve the series of events leading up to and

following the disaster in Karrat Fjord in June 2017. We show that it would not be possible to establish both timing and location of all events based on one method alone. We contextualize our results using geological knowledge of the area derived from limited fieldwork and previous studies and discuss the possible trigger mechanism.

## Study area and geological setting

The study area is located on the south facing slope of the Ummiammakku mountain in Karrat Fjord, central West Greenland (Fig. 1A). The topography is influenced by quaternary glaciations with up to 2000 m high oversteepened slopes and long fjords up to 1100 m deep. The climate is arctic with a mean annual air temperature of -3.9 at sea level in the town of Uummannaq 110 km to the south and the slopes in the region are permanently frozen (Westergaard-Nielsen et al., 2018). At present West Greenland represent an area of tectonic stability and only few minor tectonic earthquakes are known (Voss et al., 2007).

The bedrock of the Karrat region is dominated by Archean gneiss interfolded during multiple events with supracrustal rocks of the Palaeoproterozoic Karrat Group (Henderson and Pulvertaft, 1967; Sørensen and Guarnieri, 2018) (Fig. 1B). Locally around the Karrat 2017 rock avalanche, the succession consists of the Archean Umanak gneiss overlain by Palaeoproterozoic quartzite and semipelitic to pelitic schist of the Karrat Group (Mott et al., 2013). The 1:100 000 scale geological map of the area shows that the bedding of the slope surrounding the Karrat 2017 rock avalanche has a general unspecified dipslope. As the wider region is polyphase deformed the regional dipping trends shows a wide variety (Henderson and Pulvertaft, 1987). Dipslope is thus a local phenomenon mostly confined to the slope of the 2017 rock avalanche and is only recorded very locally elsewhere in the wider area. The slope is in places covered by thin colluvium and glacial erratics.

## Methods and data

We use a workflow integrating seismological data, SAR and optical imagery – all publicly available – for describing the evolution of the Karrat Landslide Complex. These data sources have different temporal and spatial resolution ranging from years to milliseconds and meters to 10's of kilometres (see Table 1). Individually they have unique information for studying unstable rock slopes, but tell an incomplete story by themselves, and the value of the individual datasets increases significantly when integrated. The workflow is preliminarily described and applied in Svennevig et al. (2019) examining a minor (ML 1.9) non tectonic seismic event in the Karrat Landslide Complex on 26 March 2018.

We found that alerting each other across disciplines of suspected smaller events enabled us to construct a reliable multi-year sequence of both confirmed smaller rock slope failures and periods of activity in the area. For example, if a seismic event was suspected of being caused by a rock slope failure, optical satellite images before and after the time of the seismic event were inspected for changes, and DInSAR images constructed for evidence of movement. Alternatively, if optical satellite images showed change between two satellite passages, we could check if a seismic event had occurred in the area in the time interval, and if DInSAR analyses showed movement to confirm either minor activity or indeed a rock slope failure.

## Fieldwork

The remoteness of the area and the steepness and elevation of the coastal slope make fieldwork logistically challenging. Because of the continued threat from rock slope failures (see below) and near constant minor rockfalls it is not safe to come closer than about 1.5 km of the scarp. These conditions highlight the need for remotely sensed data as exemplified below. However, data on the structural setting was not possible to get without field visits and for this reason we visited areas just east and west of the Karrat 2017 rock avalanche on two short reconnaissance stops during the summer of 2019. Further data was collected on a helicopter fly-by and using a camera-equipped multirotor drone.

## DInSAR

Slope deformation can be detected remotely using techniques based on Differential Synthetic Aperture Radar Interferometry (DInSAR) (Rosen et al., 2000). The main observable is a so called differential interferogram, namely a map of the phase differences between two radar images, which is confined (wrapped) within the fundamental $[-\pi \ \pi]$ interval. Providing the interferogram phase can be correctly unwrapped (Ghiglia and Pritt, 1998), the one-dimensional ground-motion between two radar acquisitions can be measured in the line-of-sight direction, i.e. towards and away from the platform carrying the radar. The evolution (time-series) of the line-of-sight deformation component can be measured with so called multi-temporal SAR interferometry techniques (Crosetto et al., 2016; Carlà et al., 2019), which are based on the generation of tens or hundreds of differential interferograms, and on the joint analysis of pixels (or groups of pixels) with a stable radar phase throughout the acquisition time span.

A prerequisite for the applicability of DInSAR is a sufficient level of statistical similarity (interferometric coherence) between the electromagnetic properties of the surface at the two acquisition times. This can be lost due to changes in the satellite viewing geometry or physical changes at the surface between acquisitions. Ground motion gradients of more than half of the radar wavelength (e.g. 2.8 cm for Sentinel-1) between acquisitions will cause a complete loss of coherence, called decorrelation, in the interferogram. In practise, decorrelation occurs already for lower ground motion rates due to other nuisance contributions to the radar phase.

Monitoring the deformation at the Karrat Landslide Complex is challenging due to several factors: rock avalanches in 2009, 2016 and 2017; high deformation rates; frequent snowfall in the winter season (October through May); steep slopes. All of the latter contribute to decorrelation in several areas and/or time intervals and limit the applicability of multi-temporal InSAR methods. In this study we discuss the application of DInSAR to imagery from the Sentinel-1A and -1B SAR satellites operated by the European Space Agency (ESA). We analysed about 180 images acquired from ascending track 90 between Oct. 2014 and Oct. 2018, and 80 images acquired from descending track 25 between July 2017 and Oct. 2018. Images were acquired every 12 days until Oct. 2016, and every 6 days after this date. The viewing geometry of track 25 is better suited for detecting motion along the slopes in our area of interest, since large parts of the slope that failed in 2017 dip steeply toward the radar in the viewing geometry of ascending track 90, causing decorrelation associated with foreshortening and layover effects (Rosen et al., 2000).

Differential interferograms between 6- and 12-day Sentinel-1 Interferometric Wide Swath (IW) Single Look Complex (SLC) products were formed using the SARPROZ software (Perissin et al., 2007), applying a 5 x 1 averaging (multi-looking) factor, resulting in an approximately 20 m x 20 m spatial resolution. The topographic contribution to the interferometric phase was removed using ArcticDEM version 2.0 (Porter et al., 2018). For interferograms following the June 2017 rock avalanche, the ArcticDEM was locally corrected with a DEM (digital elevation Model) derived from oblique photogrammetry collected in the summer of 2017 (courtesy of E.V. Sørensen, GEUS).

### Seismology

The Geological Survey of Denmark and Greenland (GEUS) monitors seismic activity in Greenland using the Greenland Ice Sheet Monitoring Network (GLISN network, www.glisn.info), which consists of 21 stations (Clinton et al., 2014). Data is screened for possible events, and manually analysed for location and magnitude using the software SEISAN (Havskov & Ottemøller, 1999). Detecting and accurately locating the activity in the Karrat area depends on having a sufficient number of nearby stations (see Fig. 1A).

Seismological data enabling us to register and locate smaller non-tectonic events became available around 2010. Until the 1990's the Greenlandic network consisted of only 3-4 stations, increasing to 5-8 stations around 2000. The present GLISN network was rolled out from 2008 to 2010 and has 21 operational stations. Before 2010 only very large rock slope failures would have been observed on the seismic stations, for example the Paatuut 2000 rock avalanche (Dahl-Jensen et al., 2004), which by luck also coincided with a temporary research network station deployment (Dahl-Jensen et al., 2003). Not only tectonic earthquakes are detected in Greenland. We see many events that we classify as non-tectonic events. This class of events were first described by Ekström et al. (2003) and were found to be located at Greenland's large outlet glaciers. The monitoring carried out by GEUS locate many non-tectonic events smaller than the globally detected events described by Ekström et al. (2003), and also many of these are located close to large outlet glaciers.

Magnitudes are calculated using the local magnitude (ML) equation for Greenland (Gregersen 1999). This equation is established for tectonic earthquakes but using it also for all types of events provides an estimate for comparison. Magnitudes of both tectonic and non-tectonic events in Central West Greenland are typically from ML 0.5 and up to ML 3.0 with a few larger. The magnitude calculated for non-tectonic events is probably too low as the low frequency content is higher than for tectonic earthquakes. The stations central West Greenland area are located along the coast with a distance of at least 100 km between them. Thus, the horizontal location uncertainty of detected earthquakes or other types of seismic events is up to 50 km, in particular in the east-west direction.

### ArcticDEM

ArcticDEM is a freely available 2 m spatial resolution digital elevation model (DEM) covering all of the Arctic area north of 60°N (Porter et al., 2018). As such, it has the highest spatial resolution of publicly available datasets covering Greenland. The DEM is derived from high-resolution (~0.5 m pixel size) stereo satellite imagery from the commercial WorldView satellites. The source images are not publicly available. Several DEM strips reflecting various image acquisition times are available covering the same areas making it possible to follow the temporal evolution of unstable

rock slopes. For the Karrat Fjord area DEM strips are available from 3 June 2008 to 23 June 2017 but in variable quality and coverage. The area of the Karrat 2017 rock avalanche is for example only partially covered by a single ArcticDEM strip from after the rock avalanche acquired on 23 June 2017.

### Space borne optical (Sentinel-2 and Landsat)

The Sentinel-2A and -2B are Earth monitoring multispectral optical satellite imaging systems operated by ESA. They record in 13 spectral bands at various resolution: four bands at 10 m (including visual light), six bands at 20 m and three bands at 60 m spatial resolution as such they are currently the highest resolution freely available optical data sets covering

Greenland. Sentinal-2A was launched in June 2015 and Sentinel-2B was launched in March 2017. Revisiting time is every five days at the equator but at higher latitudes such as Greenland most areas are covered twice or more every week. At high latitudes with constant winter darkness there is a data gap in the winter months. For this reason, there is a yearly data gap at the Karrat Landslide Complex from the end of October to beginning of March.

Landsat images were used to extent the image coverage back past the launch of the Sentinel-2 satellite (pre-June 2015).

The Landsat program is a series of Earth monitoring multispectral optical satellites, the first of which, was launched in 1972. Landsat 1-5 (1972-1993) had spatial resolutions of 60 m and Landsat 6-8, a 30 m resolution. Landsat-7 and -8 revisits the same area every eight day since the launch of Ladsat-8 in 2013. Further back in time the coverage is sparser. As is the case for Sentinel-2 scenes Landsat images at this high latitude have a winter data gap from end of October to beginning of March. Data are freely available from USGS within 24 hours of acquisition. Landsat scenes were processed

and inspected same as the Sentinel-2 images.

Scenes were downloaded freely and RGB images were produced using the freely available Sentinel Application Platform (SNAP). For the present study, we have only performed visual interpretation of the RGB Sentinel-2 images looking manually for changes on the slopes.

### Aerial images

To constrain the evolution of the Karrat Landslide Complex a set of 1:45 000 scale black and white aerial photos from 1953 (available from The Danish Agency for Data Supply and Efficiency) have been analysed. These constitute the oldest known aerial images from the area.

Table 1: Temporal and spatial resolution of the various datasets.

| Method/source | Resolution | | Period |
|---|---|---|---|
| | Spatial | Temporal | |
| Space borne InSAR (Sentinel-1) | 5x20 m | 6 days | Since 2014 |
| Seismology | 10 to100s of km | seconds | since ~2000 |
| ArcticDEM | 2 m | years/months | Variable from 2008 to 2017 |
| Optical — Space borne (Sentinel-2) | 10 m | Few days | since 2015 |
| Optical — Space borne (Landsat) | 30-60 m | Weeks - Months | Since 1973 |
| Optical — Aerial images nadir | 20-30 m | years/decades | Variable since 1953 |

## Results – evolution of the Karrat Landslide Complex

The tsunami on 17 June 2017 spurred immediate investigations of the coastal slopes in the region. The scarp of the rock avalanche in the Karrat Fjord was localised by the Danish Defence Command within hours after the event. At the same time, another area showing clear signs of deformation was noticed 500 m west of the scarp. Further investigations based on satellite imagery revealed that the 2017 rock avalanche had been preceded by smaller rock avalanches immediately to the east (e.g. Bessette-Kirton et al. 2017). This was the state of knowledge prior to this study.

In order to describe the multifaceted evolution of the Karrat area, it is necessary to establish a nomenclature framework. Hence, we introduce the Karrat Landslide Complex as a 3 by 9 km area of past, present and future rock slope activity on the south facing slope of Karrat Fjord, 30 km east of the village of Nuugaatsiaq, West Greenland (Fig. 1A). The three rock avalanches in the Karrat Landslide Complex are named according to the year they happened. The three unstable rock slopes that have not yet failed catastrophically (*sensu* Hermans and Longva 2012) are termed areas 1, 2 and 3 from west to east (Fig. 1B,C,D). Inspection of the seismological records document numerous shallow non-tectonic seismic events some of which we interpreted to be activity in the unstable rock slopes. These are named after the date they happened and seismic event, e.g. 2017-06-01 seismic event. The 2009, 2016 and 2017 rock avalanches, the three unstable rock slopes and the seismic events are described below and are listed chronologically from oldest to newest in Table 2, bearing in mind that the list is probably not complete.

### Sign of previous activity

In satellite images and aerial photos from 1953 a 0.10 km$^2$ lobe shaped feature below the area of the future scarp of the Karrat 2016 and 2017 rock avalanche is interpreted as the lobe of a minor rock slope failure modified as a rock glacier or debris flow. A conspicuous boulder field just below this feature adds to this interpretation (Fig 2A,B). The recent rock avalanches have now erased these features. East of the lobes of the recent rock avalanches, hummocky boulder fields

were observed and are here interpreted also to be older rock avalanche deposits, although the individual back scarps and lobes of these are not readily identified (Fig. 2C).

## Structural field observations

During two short reconnaissance stops the bedrock of Palaeoproterozoic metasediments on slope were observed to part easily along distinct layering of the bedding (s0 foliation) which dips 10 to 30° to the south towards the fjord. This observation is in accordance with the general dipslope described on the geological map (Henderson and Pulvertaft, 1987).

In addition, E-W orientated vertical penetrative open fractures with a normal offset are observed locally on the slope (Fig. 3).

## The rock avalanches

### The Karrat 2009 rock avalanche (71°38'20"N, 52°19'16"W, 1 September 2009 at14:09Z, ML 2.7)

The scar of the 2009 rock avalanche shows that it released along a near-vertical back scarp (Fig. 1B,D). The basal sliding plane is covered by boulders but is interpreted to follow the bedding dipping 10-30° towards the fjord (Fig. 2C). Based on this, we suggest that the 2009 rock avalanche initiated as a dipslope failure. The timing of the avalanche was initially confined to a five-year interval by the two oldest ArcticDEM strips (3 June 2008 – 12 October 2013). Google Earth Images from 1 May 2009 shows no larger recent activity and thus further constrain the event (Fig. 4A). It was then further

confined to an eight-day interval between 26 August and 2 September in 2009 by visual inspection of Landsat 7 scenes. The area appears as a 0.4 km$^2$ dark coloured patch in the latter scene. The interpretation of this patch as a rock slope failure was confirmed by inspection of the following ArcticDEM scene form 12 October 2013 and a Sentinel-2 scene from 30 July 2016 (Fig. 4B). A screening of the seismicity for the period 26 August 2009 to 2 September 2009 revealed an event on 1 September 2009 at 14:09Z as a ML 2.7 magnitude seismic event located within a 60 by 6 km EW oriented

ellipsoid. The rock avalanche was previously termed The "East landslide" by Bessette-Kirton et al. (2017) and was suggested to have occurred between 23 May 2009 and 28 April 2011 based on interpretation of Worldview images. Based on ArcticDEM strips from before (3 June 2008) and after the rock avalanche, we calculate the volume of the source area to be 2.7 x 10$^6$ m$^3$ and the lobe to be 2.8 x 10$^6$ m$^3$. That these volumes are roughly the same indicates that none of the material reached the sea and the Karrat 2009 rock avalanche is thus unlikely to have produced a tsunami. InSAR data

from 2015 show that the depositional lobe was not completely stable six years after the avalanche.

### The Karrat 2016 rock avalanche (71°38'24"N, 52°19'41"W, 15 November 2016 at 14:09Z, ML 2.1)

The 2016 Karrat rock avalanche occurred immediately west of the 2009 scarp (Figs. 1B,D, 4C)., and had the same east-west oriented vertical back scarp and dipslope weakness as surface of rupture. Due to the constant winter darkness the

255 timing of the rock avalanche, based on Sentinel-2 data alone, could only be loosely constrained to some time during the winter of 2016-2017. However, a DInSAR interferogram formed from Sentinel-1 images acquired on 11 and 17 November 2016 show a localized loss of coherence, compatible with a rock avalanche in this area (Fig. 5). Analysis of

the seismic signal reveals that a magnitude ML 2.1 non tectonic event took place on 15 November 2016 at 11:34Z (Fig. 6). The westernmost part of the scarp is visible in an ArcticDEM strip from 5 June 2017. Based on this DEM and the geometric constraints of the scarp of the Karrat 2009 rock avalanche, we calculate the volume of the source area to be 3.0 x $10^6$ m³. It is not possible to constrain the volume of the deposit from the rock avalanche as no DEM covers the entire area. Bessette-Kirton et al. (2017) described an enlargement of their "East landslide" (here named the Karrat 2009 rock avalanche) that took place sometime between 16 May 2016 and 5 June 2017, based on Worldview images. This probably corresponds to the Karrat 2016 rock avalanche.

### The Karrat 2017 rock avalanche (71°38'36"N, 52°20'12"W, 17 June 2017 at 23:39Z, M$_s$ (20 sec) 4.2)

The landside rock avalanche of 17 June 2017 appears to have initiated as a dipslope failure. This is based on the same criteria as the Karrat 2009 and 2016 rock avalanches: dipslope of the bedrock on the coast and near vertical east-west oriented back scarp. The Karrat 2017 rock avalanche is documented in all of our data sources, but these are secondary to the eye witness reports of the landslide rock avalanche and tsunami that combined with the seismic signal gives the exact timing of the event to 17 June 2017 at 23:39Z (Fig 6A). The Karrat 2017 rock avalanche is described in previous preliminary publications (Bessette-Kirton et al., 2017; Clinton et al., 2017; Gauthier et al., 2018). It was termed the "Nuugaatsiaq landslide" by Bessette-Kirton et al. (2017) and Poli (2017) after the village of Nuugaatsiaq 30 km to the West and the "Greenland landslide" by Chao et al. (2018). Only the easternmost part of the rock avalanche is visible in two ArcticDEM strips covering the area from 23 and 28 June 2017. Previous volume estimates range from 35 to 76 x $10^6$ m³ (Bessette-Kirton et al., 2017; Chao et al., 2018; Gauthier et al., 2018; Paris et al., 2019) but some of these are based on DEM work that does not include the full volume of the Karrat 2016 rock avalanche, see discussion. Recent work based on detailed DEMs from high resolution oblique photogrammetry from 2015 and 2017 gives a volume of 41 – 43.5 x $10^6$ m³ mobilized in the 2016 and 2017 rock avalanches (Sørensen et al in prepXX). Subtracting the volume of the Karat 2016 rock avalanche given above gives a volume of 38-40 $10^6$ m³ for the Karrat 2017 rock avalanche.

### The unstable rock slopes

#### Area 1 (71°38'44"N, 52°28'19"W)

Area 1 is a very large and well-developed unstable rock slope, 4 km west of the present Karrat 2017 scar (Fig. 1B,C) which is not previously described in the literature. The 2000 m by 1600 m area is defined by a well-developed up to 120 m high back scarp and lateral release surfaces. The back scarp is near the crest of a 1000 m high mountain and the unstable area extends to the coast, suggesting that it continues below sea level. Internally, the unstable slope shows signs of significant strain with multiple scarps, contour parallel grabens and an overall hummocky fabric. The area is well defined in the 1953 aerial images with a well-developed backscarp and a hummocky morphology indicating that it had already undergone significant internal strain at that time (Fig. 2A). Whether the area was active or dormant at this time is unclear. Subareas in the lower 200 to 400 m of the slope show downslope movements between various ArcticDEM scenes. The same areas decorrelated in almost all DInSAR interferograms (Fig. 5). Some interferograms show episodic movement of 1-5 mm/day over most of the area (Fig. 5D). Multiplying the height of the back scarp with the area of the unstable slopes (120m x 2000m x 1600m) gives a tentative minimum volume of 380 x $10^6$ m³ for the area above sea level.

Area 2 (71°38'46"N, 52°21'57"W)

Area 2 is a 500 by 700 m well-developed unstable rock slope, located 500 m west of the Karrat 2017 rock avalanche at 950-1200 m elevation (Fig. 1B,D). The area could not be visited during fieldwork due to the steepness of the terrain and the near constant rock falls. Drone inspection of the area showed that it is covered by thick colluvium, however, the Proterozoic bedrock is exposed in the 50 m high back scarp demonstrating that the instability most likely involves bedrock (Fig. 7A). The area appears as a bulge in the oldest ArcticDEM strip (3 June 2008) indicating that activity in the area could be older than this. However, it is not possible to identify the onset of activity using either Landsat or older aerial images due to their coarse spatial and temporal resolution. Bessette-Kirton et al. (2017) call the area the "West landslide" and propose movement started between 13 May 2015 and 16 May 2016, based on WorldView imagery. This is in accordance with InSAR analysis showing the first subtle signs of deformation in the area as a loss of coherence during 3 May 2015 and 15 May 2015. We interpret a ML 1.8 seismic event on 13 May 2015 at 17:14Z (Table 2) from the area to represent the exact time of the first major displacement of the area. Deformation of the outer boundary of the area is clearly visible in all InSAR images from end of May 2015 with movement on the order of 1 mm/day (Fig. 5A). After September 2015 and up to the present day the entire area shows loss of coherence in InSAR which can be due to fast motion or change in surface properties, both of which suggest an acceleration in activity (Fig. 5B-D). This is in agreement with the very broken up fabric observed in the field (Fig. 7A), indicating both fast movement and change of surface properties. Assuming that the height of the back scarp represents the minimum average thickness of the unstable mass, we tentatively model the volume of Area 2 to be at least $13 \times 10^6$ m$^3$ (by multiplying the area of 260 000 m$^2$ by an average thickness 50 m). Paris et al. (2019) used volumes between $2 \times 10^6$ and $38 \times 10^6$ m$^3$ for the area for tsunami modelling mentioning that the $38 \times 10^6$ m$^3$ is the more realistic estimate.

Area 3 (71°38'32"N, 52°21'23"W)

Area 3 is an 800 by 500 m unstable rock slope located between the scarp of the Karrat 2017 rock avalanche and Area 2 (Fig 1B,D). A clear backscarp is not visible but the area shows signs of deformation since May 2015 (coinciding with the initiation of Area 2) and decorrelation over the entire area in all interferograms since 21 June 2017 (the first acquisition after the 17 June 2017 rock avalanche) (Fig. 5). Localized rockfalls are seen in Sentinel-2 images and during the field visit. It has an overall hummocky surface in recent ArcticDEM strips and a broken up internal fabric was observed in the field (Fig. 7B) indicating significant internal strain. Upslope the area seems to be defined by the western continuation of the backscarp of the Karrat 2017 rock avalanche and it is reasonable to assume that it is sliding on the same basal sliding plane. This area is described here for the first time. We infer activity in Area 3 to have started in May 2015 and increased considerably after 17 June 2017 as the block dislocated in the Karrat 2017 rock avalanche would have supported the area and prevented it from moving. The volume is constrained by using the western continuation of the back scarp and the basal sliding plane of the Karrat 2017 rock avalanche. The western extent of the area is confined by the observed movement in InSAR. This gives a tentative volume of $11 \times 10^6$ m$^3$.

Non-tectonic seismic events

The causes of non-tectonic events are several. For example, events with epicentre located near an outlet glacier (cryo-seismic events) often contain a low frequency component, and are usually much longer in duration than tectonic

earthquakes (Fig. 6), and are interpreted to be caused by calving of glaciers (Ekström et al., 2003; Nettles et al., 2008). Other non-tectonic events, in Western Greenland, are mainly caused by sea ice breakup, glacier or sea ice movements on bedrock, but other types are also present see e.g. Podolskiy and Walter (2016). Rapid rock slope failures also produce a seismic signal. The Karrat 2017 rock avalanche was seen globally as a Ms 4.2 event (U.S. Geological Survey, 2020), and the 2000 Paatuut rock avalanche was seen throughout Greenland (Dahl-Jensen et al., 2004). Smaller events associated with known rock slope failures (this paper) are only seen more locally (Fig. 6).

We have chosen to show the data in Figure 6 without filtering. Although we use different bandpass filters when analysing the data for location, the very different frequency content of the different events (tectonic, glacial and rock slope failures) are best seen with no filter, highlighting the differences.

Non-tectonic events can easily be identified from tectonic earthquakes based on their different frequency content and P and S amplitudes (Fig. 6).

However, distinguishing a rock slope failure signal from other non-tectonic events, such as events associated with glaciers, is not straightforward. The seismic signature from two very large rock slope failure events, the 2000 Paatuut landslide (Dahl-Jensen et al., 2004) and the 2017 Karrat rock avalanche have long lasting tremor signals and a strong low frequency component. For smaller rock slope failures the tremor component will be smaller in duration and amplitude. Many aspects of smaller known rock slope failures are similar to cryo-seismic events (Fig. 6).

From the analysis in this paper we have built an experience database of the seismic signature of rock slope failure. We have analysed events from the Karrat area, using both the location of events, the seismic signature and the evidence from optical and InSAR satellite data to distinguish the types of events. The seismic signal from the major Karrat 2017 rock avalanche is also clearly not a tectonic event – there is no P wave arrival and only a very low frequency S arrival. However, the cryogenic seismic events and smaller rock slope failures have many characteristics in common. They have a longer duration, a lower frequency content, and often no or very unclear P arrivals. The geographical observation that several large outlet glaciers are found in the area around the Karrat Landslide Complex makes it necessary to look deeper into the characteristics of these non-tectonic events. We have looked at the time difference between P and S arrivals at the Nuugaatsiaq seismic station (when possible). This time difference can be translated into a distance using an earth model with the P and S wave velocities. If the distance from Nuugaatsiaq matched the distance to the Karrat area, it is an indication that it might be a seismic event associated to the rock slope. However, there are several large outlet glaciers within 30 – 60 km of Nuugaatsiaq, and with the uncertainty in location up to 50 km, the time difference is not a conclusive parameter. We have also looked at the duration of the events. Typically, the cryogenic seismic events have a duration of several minutes, while the known and suspected rock slope failures are shorter – from 45 s to 90 s. But there are also suspected cryogenic seismic events that are of the same duration as suspected rock slope failures. Currently, we must rely on supporting evidence from the satellite data in order to confirm or dismiss a suspected rock slope failure seen seismically.

Several seismic events have been tied to activity in the Karrat Landslide Complex occurring both before and after the Karrat 2017 rock avalanche. Svennevig et al. (2019) described a seismic event from the 26 March 2018 and suggested it was related to rock slope activity based on observed rockfall in Sentinel-2 images before and after the event. Several

seismic events during the period from 2009 and to the time of submission are suspected to be associated with rock slope failures and are listed chronologically in Table 2. Another example is the ML 1.9 non-tectonic seismic event that occurred on 1 June 2017 at 20:55Z in the area, with an S-P phase arrival time difference at the seismic station NUUG corresponding to the distance between Nuugaatsiaq and the Karrat Landslide Complex area. The seismic signal is similar to those of the Karrat 2009, 2016 and 2017 rock avalanches (Fig. 6) but the event could not be confirmed by InSAR and optical interpretation due to poor data coverage in the short period between the event and the later Karrat 2017 rock avalanche. It is thus reported here as a seismic event that is possibly a small rock slope failure.

A denser local seismograph network in central West Greenland was rolled out during the summer of 2019. This will improve the location accuracy of events in the area – including the Karrat Landslide Complex – allowing event location to help separate non-tectonic events into cryogenic seismic events and possible rock slope failures.

The events up until the time of submission are listed in Table 2.

Table 2: Summary of chronological listing of events at the Karrat Landslide Complex

| Event | Timing | Note | Evidence/data source | | | |
|---|---|---|---|---|---|---|
| | | | Seismic (ML) | Optical | DEM | DInSAR |
| Area 1 initiates | Pre-1953 | Well-developed scarp visible in legacy GEUS aerial images (1953), present-day deformation confirmed by DInSAR. | | X | | X |
| Karrat 2009 rock avalanche | 2009-09-01T14:09Z | First recent rock avalanche. | 2,7 | X | X | |
| Seismic event | 2014-09-19T04:30Z | Seismic signature of a rock slope failure. | 2.0 | | | |
| Area 2 and 3 initiates | 2015-05-13T17:14Z | Deformation only in parts of area 3, localised deformation in area of 2017 rock avalanche. | 1.8 | | X | X |
| Karrat 2016 rock avalanche | 2016-11-15T11:34Z | Second recent rock avalanche. | 2.1 | X | X | X |
| Seismic event | 2017-06-01T20:55Z | Seismic signature of a rock slope failure, but not resolved by other datasets. | 1.9 | | | |
| Karrat 2017 rock avalanche | 2017-06-17T23:39Z | Third recent rock avalanche. Described by Bessette-Kirton et al. (2017) and Gauthier et al. (2018) and eye witnesses. | 4.2 | X | X | X |
| Seismic event | 2018-02-21T01:10Z | Seismic signature of a rock slope failure, but not resolved by other datasets. | 1.7 | | | |
| Seismic event | 2018-03-26T21:21Z | Described by Svennevig et al. (2019). | 1.9 | X | | |
| Seismic event | 2018-04-19T20:18Z | Several consecutive seismic events over a period of two hours. Interpreted to be rock slope failures. | 1.9 | | | |
| Seismic event | 2018-08-13T10:04Z | Seismic signature of a rock slope failure. Movement several places in the Karrat Landslide Complex including large parts of Area 2 seen in all other datasets. | 1.2 | X | X | X |
| Seismic event | 2018-08-17T01:15Z and 01:18Z | Seismic signature of a rock slope failure , but not resolved by other datasets. | 1.7 | | | |
| Seismic event | 2019-10-08T01:50Z | Seismic signature of a rock slope failure , but not resolved by other datasets. | 0.9 | | | |

# Discussion

## Evaluation of the workflow

This study shows the effectiveness of combining complementary remote sensing techniques to establish precise time and location of a series of successive rock slope failures in a remote Arctic setting where fieldwork is challenging due to limited infrastructure. It is an inexpensive setup relying on freely available and continuously updated datasets. However, some unstable slopes may go undetected due to the inherent limitations of the remote sensing data, such as the lack of optical data during winter season, the resolution problems of DInSAR in steep terrain, and the location errors of seismic events. Questions regarding the development of an unstable slope prior to failure, the triggering mechanisms, and the type of failure, may be only partly resolved by remote sensing alone. Furthermore, a reliable assessment of possible failure scenarios and their associated hazards requires validation by structural mapping and displacement measurements in the field (e.g. Oppikofer et al., 2013; Hermanns et al., 2016).

The methodology demonstrated here might also be useful in other, less remote settings, by providing a means of monitoring the activity of a known unstable rock slope. Once the seismological, InSAR and optical signatures of a rock slope failure are established, the workflow can be used to detect and locate rock slope activity and thus focus fieldwork. Semi-automatic methods have been developed to detect landslides based on either satellite optical or SAR data (e.g., Martha et al., 2010; Friedl and Hölbling, 2015), and the detection capability of such methods may be improved by combining the different data sources as indicated by the results of our study.

## Evolution of the Karrat Landslide Complex

As our compilation of results shows, the Karrat 2017 rock avalanche was not an isolated event, but part of an ongoing process of successive rock slope failures focused in the Karrat Landslide Complex, (Figs. 1, 8). The activity can be subdivided into one or more previous phases (prior to 2009, Fig. 3) and a recent phase initiating with the 2009 rock avalanche and so far culminating with the 2017 rock avalanche (Figs. 4, 5, 6).

Area 1, the large unstable rock slope in the western part of the Karrat Landslide Complex, probably had a long history of activity as indicated by the well-developed backscarp and hummocky morphology in the oldest aerial photograph of the area from 1953 (Fig. 2A). The boulder fields and hummocky lobes in the eastern part of the complex, here interpreted to be deposits from older rock avalanches (Fig. 2B,C), furthermore points to a previous stage of activity in the Karrat Landslide Complex, possibly several 100s or 1000s years old but younger than the previous glaciation as this would presumably have erased the morphological expression of the rock avalanches.

The 2009 rock avalanche was the earliest detected seismological event as the first GLISN stations came into operation during the summer of 2009. However, a less dense seismograph network was present prior to this (back to 2000) and records no larger events in the area. After the 2009 rock avalanche, no activity was recorded in the Karrat Landslide Complex for another five years until a seismic event with a signature of a rock slope failure was recorded on 19 September 2014 (see Table 2). The period has reasonable optical satellite and seismological coverage, indicating that the lack of recorded events reflects a real hiatus in activity. Activity picked up after the 2014 event as the two unstable areas termed

area 2 and 3 started to show signs of deformation in InSAR and optical imagery from May 2015 (Fig. 5A), followed by the Karrat 2016 rock avalanche, a seismic event with signature of a rock slope failure on 1 June 2017, and culminating with the major rock avalanche on 17 June 2017. A number of seismic events interpreted to activity in the Karrat Landslide Complex were recorded during 2018 and 2019, of which some correlated with rock slope deformation observed in all other considered datasets 2018 (e.g. Svennevig et al., 2019). These events show that the unstable slopes in the Karrat Landslide Complex continue to be active and may pose a continued threat of catastrophic failure.

We cannot conclude whether the three rock avalanches were preceded by precursory deformation, due to the rather coarse resolution of the optical satellite imagery and the problems in the ascending InSAR data due to steep topography. However, the east-to-west migration of the rock avalanches in the eastern part of the Karrat Landslide Complex suggests a westward migration of a fracture system acting as back scarps for the three recent rock avalanches. Multiple rock slope failures from the same site are well known in the literature, and it has previously been shown in mountain areas of Europe how previous massive rock slope failures can increase the likelihood of new rock slope failures (Hermanns et al., 2006). The east-west migration of rock slope failures, along with the ongoing deformation detected by InSAR in Area 2 and 3 (Fig. 5D), point to the area just west of the Karrat 2017 rock avalanche as the most likely area of future catastrophic failure. The relationship of Area 1 to the other parts of the Karrat Landslide Complex is ambiguous. The trend of the bedding (s0 foliation) acting as a dipslope basal surface of rupture of the Karrat 2009, 2016 and 2017 rock avalanches seems to be situated just below sea level to the west below Area 1 and might act as the basal sliding surface of this also (Fig. 1B).

## Preconditioning and preparatory factors for slope instability

Based on the remotely sensed data and our limited fieldwork, it has not been possible to determine exactly what factors (*sensu* Glade and Crozier, 2005) triggered individual rock slope failures and seismic events in the Karrat Landslide Complex and why there seems to be a recent peak in activity since 2009. The structural setting most likely is a precondition factor as the Karrat Landslide Complex coincides with an area of weak bedding (s0) foliation, local dipslope and coastal parallel vertical jointing (Fig. 3). This does, however, only suggest the where but not the when. We observe that events may occur throughout a year and, from the limited data we have available, no seasonal change in activity can be seen (Fig. 8C). This indicates that something with a longer period than the seasonal cycle could contribute as a preparatory factor.

Regional models propose that slopes in this part of west Greenland are permafrozen (Westergaard-Nielsen et al., 2018), however, the specific permafrost state of slope at the Karrat Landslide Complex is not known. The regional air temperature has increased by 4-5 °C since 1880 and this increase has been accelerating since c. 1990 (Cappelen et al., 2018) making it reasonable to speculate that this could have an effect on the permafrost conditions. It is well known that permafrost degradation can play an important role in slope stability (Draebing et al., 2014; Krautblatter et al., 2013). We therefore hypothesize that permafrost degradation may be the main preparatory factor for the recent slope instability and rock avalanches. With the projected temperature increase of up to 8 °C towards 2100 (IPCC, 2013), a range of preparatory factors is expected to change, including permafrost degradation and thus the likelihood of more rock slope failures from the Karrat Landslide Complex could be expected to increase.

Several works have been done on rock slope failures in deglaciated mountain settings (e.g. Norway, Böhme et al., 2015; Hilger et al., 2018) where peaks in previous activity has been dated and suggested to reflect glacial debuttressing and climatic change in the form of change in regional precipitation patterns (amount and type) and increase in temperature leading to permafrost degradation. It is, however, unclear whether these conditions translate to the much cooler high arctic conditions of West Greenland and more work is needed on this.

A variety of methods could be applied to examine this, such as dating the older (Holocene) rock slope failures, analysing aerial images from the past century to constrain more recent evolution and installing climate sensors to constrain the present permafrost conditions of the slope (Magnin et al., 2019). Bathymetrical studies of the seabed to map past rock avalanche deposits off the Karrat Landslide Complex could also be included.

## Regional hazard evaluation and context

As a whole, the occurrence of rock slope failures in the Karrat area (geological area of Proterozoic metasediments interfolded with Archean gneiss: Fig. 1) is not particularly higher than elsewhere in Greenland (Svennevig, 2019). In this context, the Karrat Landslide Complex is a local entity probably preconditioned by local dipslope. As such it is an outlier with respect to unstable rock slopes as neighbouring slopes in the fjord system with similar types of bedrock (but no 470 dipslope) show no abnormal rock slope activity. The regional landslide hazard in the Karrat area, with the exception of the Karrat Landslide Complex, is thus not thought to be higher than elsewhere in Greenland. Local occurrences of dipslope in the region should however be examined in more detail.

A consequence of the multistage evolution of the Karrat Landslide Complex is that Gauthier et al. (2018) and Paris et al. 475 (2019) overestimated the volume of the Karrat 2017 rock avalanche. Gauthier et al. (2018) using an ArcticDEM strip from May 2015 and Paris et al. (2019) using a Spot6 stereoscopic image acquired on 22 July 2013 both estimated 45 x $10^6$ m$^3$ of material effectively reached the sea. Both of these estimates include DEMs from before the Karrat 2016 rock avalanche, and thus include this volume in their estimate of the total volume that failed on 17 June 2017. Thus the tsunami run-up estimates by Paris et al. (2019) using the overestimated volumes may be taken as minimum estimates as the 480 volumes used to train the tsunami model was overestimated. Bessette-Kirton et al. (2017) used a DEM from satellite images collected on 6 May 2017 and thus do not include the volume of the 2016 rock avalanche in their volume estimate of minimum 33.4 x $10^6$ m$^3$. A detailed evaluation of volumes of the Karrat 2017 rock avalanche based on oblique photos from before and after the 2017 rock avalanche is under way (Sørensen et al in prepXX).

## Conclusions

This study shows the effectiveness of using the multi-disciplinary approach here described for studying unstable rock slopes in remote Arctic areas with difficult fieldwork conditions. This is demonstrated through the recognition of three unstable slopes and the potential of workflow to describe the evolution of rock slope failures in the Karrat Landslide complex. Due to the inherent limitations described above, remote sensing data alone cannot provide basis for detailed 490 analysis or forecasting of future rock slope failures. However, we have learned that being alert to smaller events in a

known unstable slope might be crucial for assessing the hazard of large, tsunamigenic rock slope failures. Additionally, by establishing the seismic, InSAR, and optical signatures of precursors for rock avalanches, it is possible to be alerted of new possible events, both in the Karrat area and elsewhere in isolated Arctic areas.

We show that the disastrous Karrat 2017 rock avalanche was not an isolated event. Smaller rock slope failures had taken place in the years preceding the major event and the area continues to be active. Recent rock avalanches took place in 2009, 2016 and 2017 and an increasing number of seismological events interpreted as small rock slope failures occurred from 2014 onwards. There is also evidence of older activity in the Karrat Landslide Complex. The Karrat Landslide Complex continues to be very active after the Karrat 2017 rock avalanche and specifically three unstable rock slopes named Area 1, 2 and 3 may pose a future hazard to people in the region and the consequence of a tsunami from these should be addressed.

The distribution of the events over the annual cycle indicates that there is no seasonality indicating that preparatory and triggering factors working on a longer cycle could be at play. We hypothesize that the slope instability is caused by permafrost degradation. However, further research is needed in order to confirm this.

## Data availability

All Greenland seismological data are freely available at GEOFON data centre of the GFZ German Research Centre for Geosciences and IRIS Data Services. Sentinel 1 and 2 data are available through the European Earth Observation "Copernicus" program. Landsat Images are available through the USGS Earthexplorer.

## Author contributions

The manuscript was written by Kristian Svennevig with significant contributions from Trine Dahl-Jensen and Marie Keiding. Kristian Svennevig carried out interpretation of optical data and data integration. John Peter Merryman Boncori, Sara Salehi, Anne M. Solgaard and Marie Keiding carried out processing and interpretation of Sentinel-1 data and wrote the method section on this. Trine Dahl-Jensen wrote the sections on seismology. Trine Dahl-Jensen, Tine B. Larsen and Peter H. Voss carried out processing and interpretation of seismic data. All authors reviewed and approved the manuscript.

## Acknowledgements

The governments of Denmark and Greenland funded the "Screening analyses of the risk for serious landslides in Greenland" in 2018 for which the original technical work was undertaken. The facilities of IRIS Data Services, and specifically the IRIS Data Management Center are funded through the Seismological Facilities for the Advancement of Geoscience and EarthScope (SAGE) Proposal of the National Science Foundation under Cooperative Agreement EAR-1261681. We acknowledge the European Union Copernicus Program and ESA for the use of Sentinel-1 and 2 data. Landsat images courtesy of the U.S. Geological Survey. The paper is published under permission of GEUS. Seismological

data collected in the Disco Bay under the INTAROS project was used in the landslide screening. INTAROS has received

funding from the European Union's Horizon 2020 research and innovation programme under grant agreement No 727890. Professor John R Hopper is thanked for constructive comments to this manuscript. The manuscript benefitted vastly from the comments of four anonymous reviewers.

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

## Figures

**Figure 1:**

Setting of the Karrat Landslide Complex. A: Simplified geological map of the region based on Henriksen et al. (2009) showing nearby seismic stations (NUUG and UMMG), prehistoric landslides (Svennevig, 2019) and the area of Fig. 1B. B: The Karrat Landslide Complex shown on a Sentinel-2 RGB image from 20 April 2019 where the coastal slope has a light snow cover emphasising the structures. Transparent polygons are the three rock avalanches (2009, 2016, and 2017) and the three unstable slopes (1, 2, and 3). The stippled lines at the rock avalanches are the extent of the individual scarps. Notice the recent rockfall from Area 2 (dark stria south of the area). Positions, name (letter) and direction of field photos in this and subsequent figures is indicated with arrows. C: Oblique helicopter photo of Area 1. From the shore to the top of the backscarp is 1000 m and the backscarp is up top 120 m high for scale. D: Oblique helicopter photo of Area 2 and 3 and the three rock avalanches (2009, 2016, 2017). Notice the hummocky morphology of Area 2 and 3 and the dust cloud east of area 3. Area 2 is 1200 m across for scale.

**Figure 2:**

Previous activity in the Karrat Landslide Complex. A: 1953 1:45 000 scale aerial photograph of the Karrat Landslide Complex showing the well-developed state of the unstable rock slope; Area 1 to the W. The three stippled lines to the

east show the positions of the scarps of the future rock avalanches (colours match Fig. 1B). B: Details of A with the stippled lines indicating the positions of the scarps of the future rock avalanches. Notice the lobate morphology (X) and boulder field (Y) indicating older rock slope failures along with the hummocky topography to the E (Z). C: Photomosaic of field photos from the summer of 2019 taken just E of the 2009 rock avalanche. The scarp and lobe of the 2009 rock avalanche is indicated with a stippled yellow line. The lobate hummocky topography interpreted to be deposits of past rock slope failures are indicated with (Z). Red helicopter for scale.

**Figure 3:**

Field photo of the bedrock geology of the coastal slope 1.5 km west of Area 1. View is towards the west. Sub vertical jointing and S0 foliation dipping 20-30° towards the fjord are prominent. The geologist is standing next to a vertical open fracture with a small normal offset (20-30 cm) apparent on the surface that could be a model for the development of the backscarp of the rock avalanches. Photo by Simon Mose Thaarup, GEUS.

**Figure 4:**

Optical satellite images of the eastern part of the Karrat Landslide Complex, the locus of recent rock avalanches. A: Scene from 1 May 2009 before the rock avalanche, a presumed older rock avalanche deposit (lobe) is marked with X, same as Fig. 1B (from ©Google Earth, image credit: Maxar Technologies). B: Sentienl-2 image from 5 April 2016 showing the situation after the Karrat 2009 rock avalanche of 1 September 2009 at 14:09Z. C: Sentienl-2 image from 1 March 2017 showing the situation after the Karrat 2016 rock avalanche of 11 November 2016 at 14:09Z. D: Sentienl-2 image from 10 April 2018 showing the situation after the Karrat 2017 rock avalanche of 17 June 2017 at 23:39Z. Scarp (red) of the 2017, 2016 and 2009 rock avalanches and depositional lobes (orange) is shown with bold lines.

**Figure 5**

Two-pass interferograms of the eastern part of the Karrat Landslide Complex. The colours denote radar phase differences in the fundamental $[-\pi \; \pi]$ interval, where one full interval of $2\pi$ radians corresponds to 5,6 cm of displacement. A: Ascending track 90 during 26 June – 8 July 2015. The deformation in the broader part of Area 1 is not clearly seen in this viewing geometry, however, two subareas with decorrelation due to high deformation rates are apparent. Deformation in Area 2 and 3 and below the 2017 rock avalanche is visible, but the steep slope itself is in layover due to the geometry of the satellite acquisition. B: Ascending track 90 during 12-24 September 2015. Area 2 shows decorrelation indicating acceleration of deformation rates. C: Ascending track 90 during 11-17 November 2016, spanning the 2016 rock avalanche, which shows up as completely decorrelated. D: Descending track 25 during 20 July – 11 August 2018, a year after the 2017 rock avalanche. The area of the 2017 avalanche shows partly coherent and incoherent deformation. Area 2 and 3 both show complete decorrelation due to high deformation rates. The upper right part of the interferograms shows varying coherent phase differences and decorrelation due to rapid movements of ice glaciers.

**Figure 6**

Seismic signatures of major events. All figures show the unfiltered data of the vertical component from the seismic station at Nuugaatsiaq (NUUG). A) are spectral plots of an tectonic earthquake (top), a cryo-seismic event (middle) and a confirmed rock avalanche (bottom). The difference in frequencies and duration between the tectonic event and the non-tectonic events is clear, while the difference between the two non-tectonic events (cryo-seismic and rock avalanche) is

more ambiguous. B) is a 5 min extract of the Karrat 2017 rock avalanche. C) - G) are 1 min 10 sec extracts for possible and confirmed rock avalanche events from NUUG. C) and D) are known rock avalanches (Karrat 2016 and Karrat 2017), and E) – G) are interpreted as possible rock slope failures in the Karrat area, but this is not supported by the other datasets.

**Figure 7**

Drone field photos from the Karrat Landslide Complex in the summer of 2019. A: Drone photo of the backscarp of Area 2. The arrow points to where bedrock is exposed indicating that the unstable rock slope is not a superficial feature. B: Drone photo of the backscarp of the Karrat 2017 rock avalanche looking towards the NW. The mottled interior of Area 3 is apparent along with the bulging nature of Area 2 in the background. See Fig. 1B for locations.

**Figure 8**

A: Data coverage, B: timeline of recent events in the Karrat Landside Complex and C: yearly distribution of rock slope failures and seismic events.

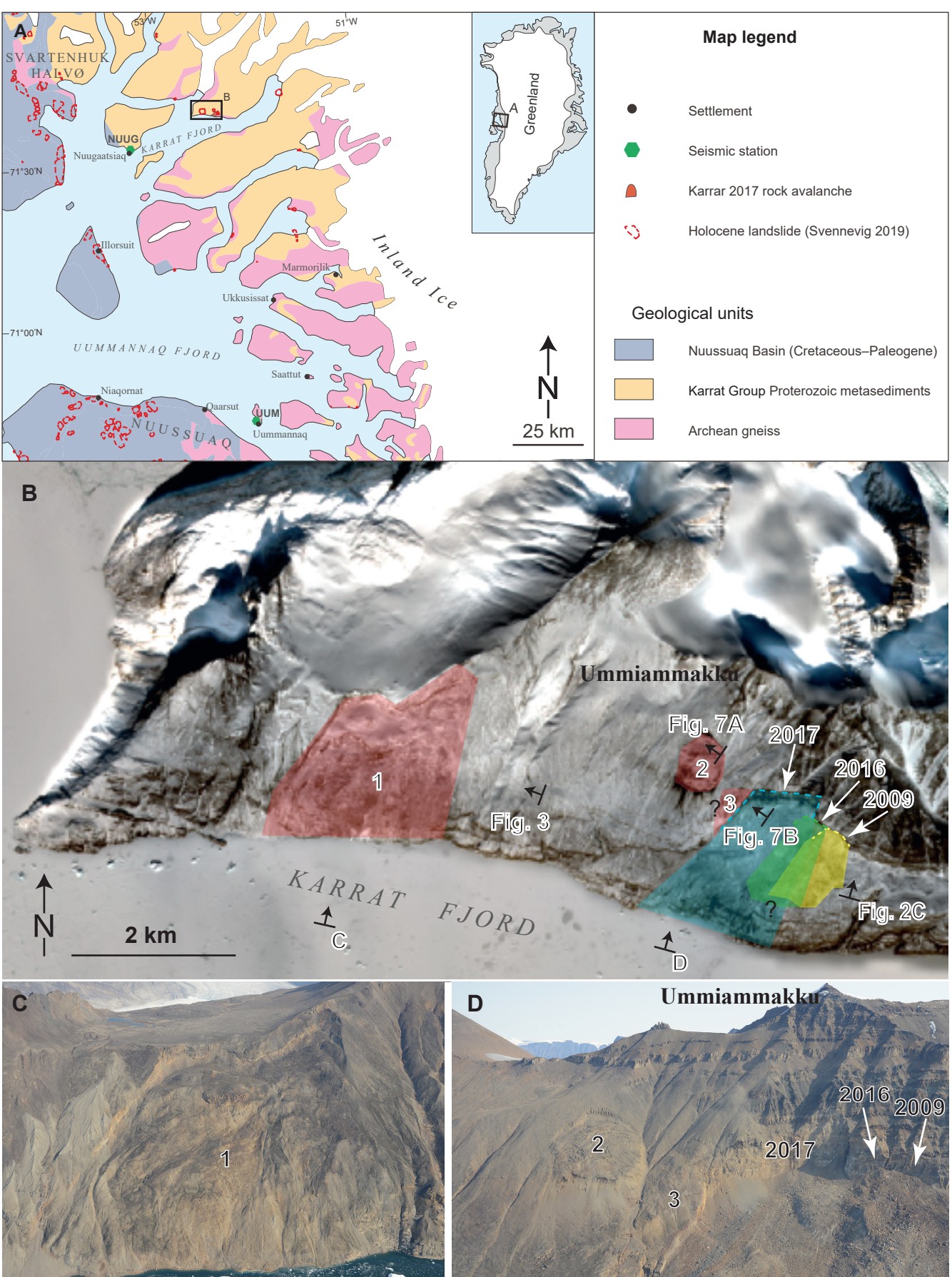

**Fig 1**

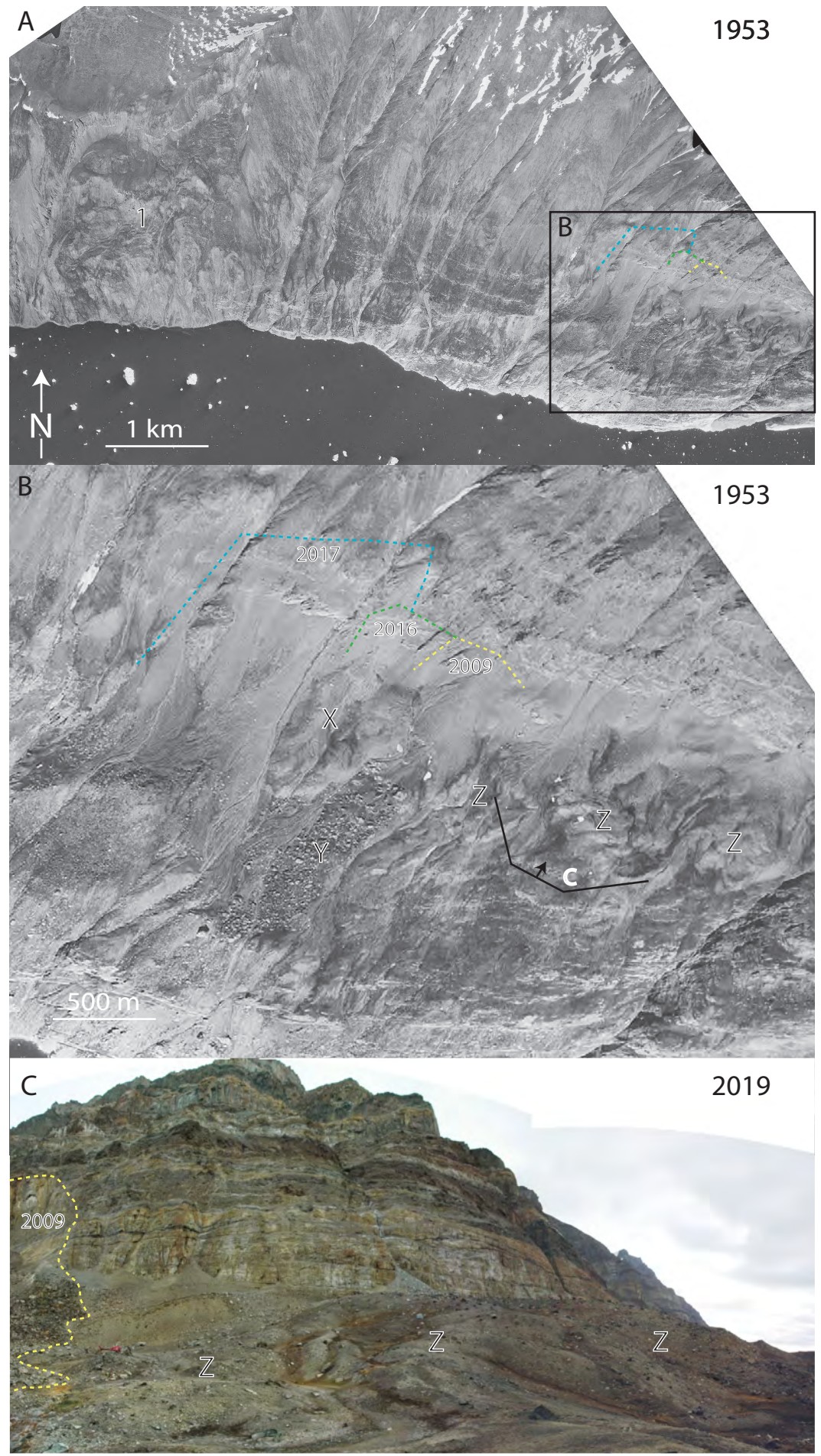

Fig. 2

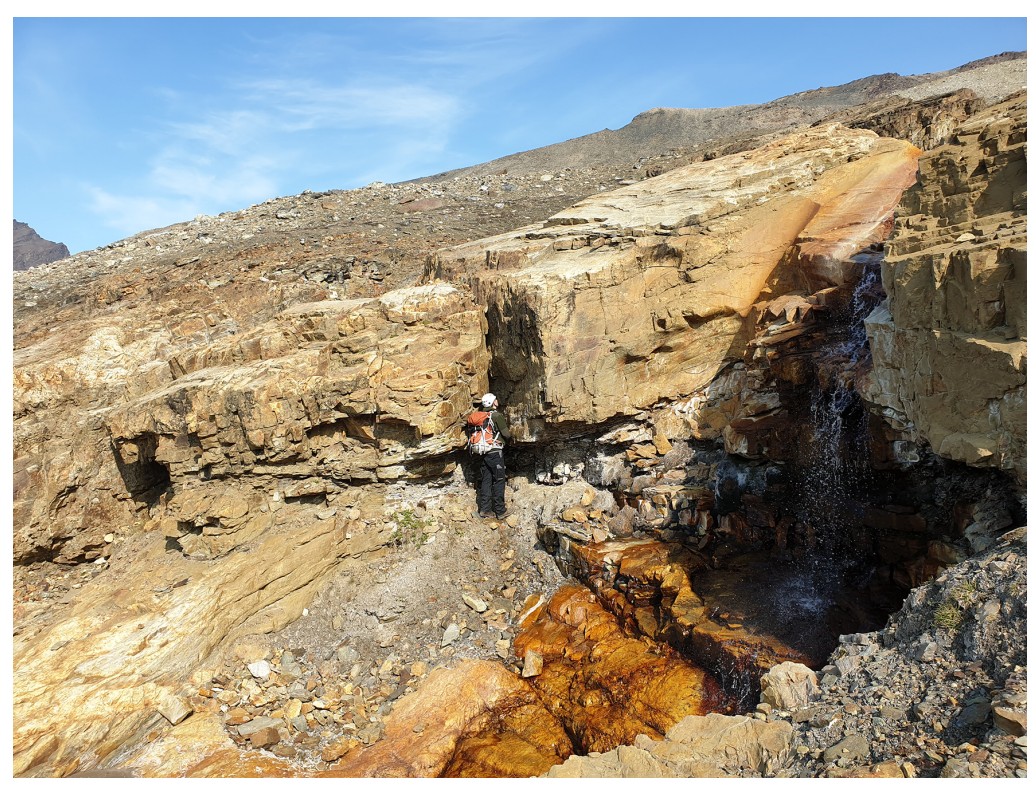

**Fig. 3**

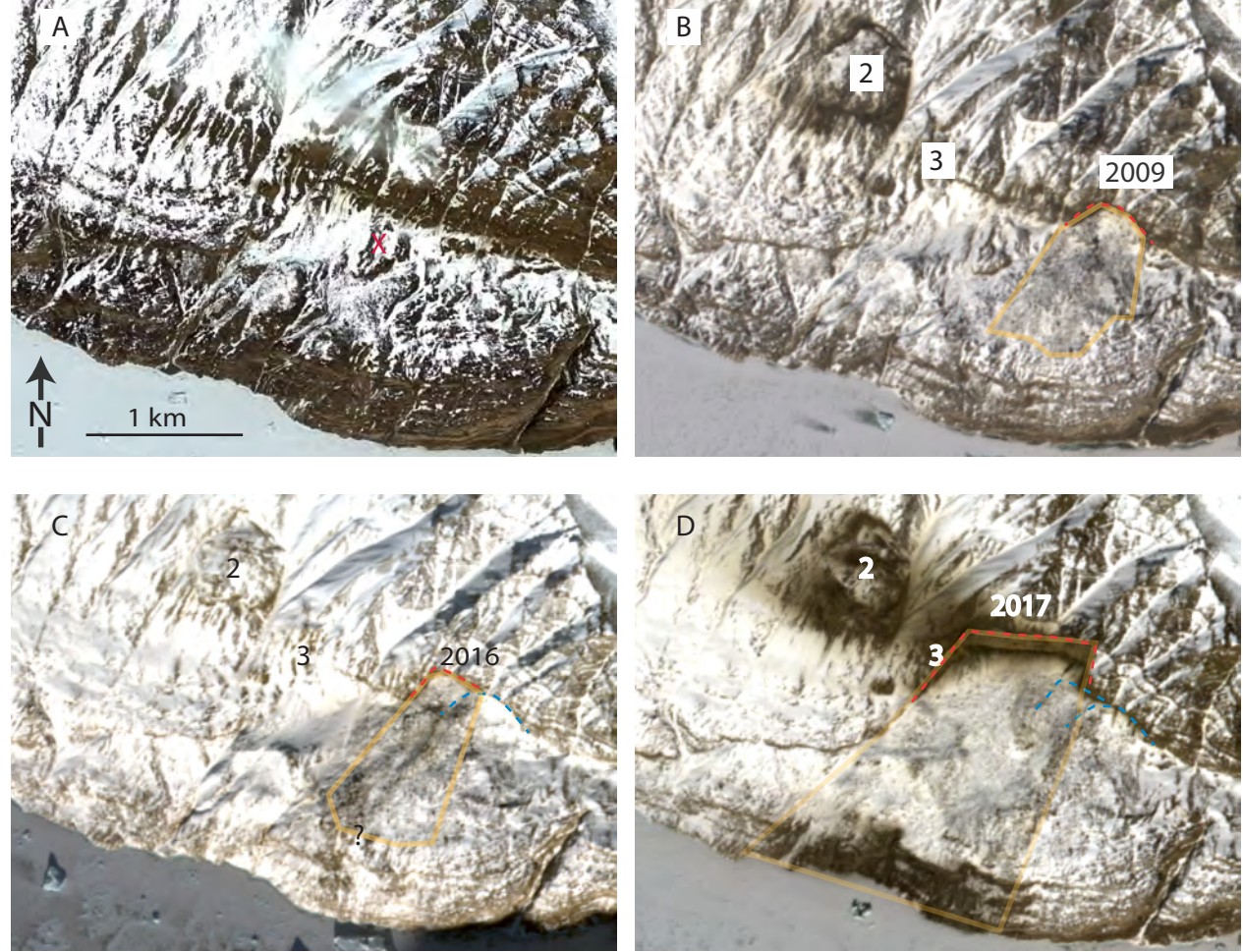

Fig. 4

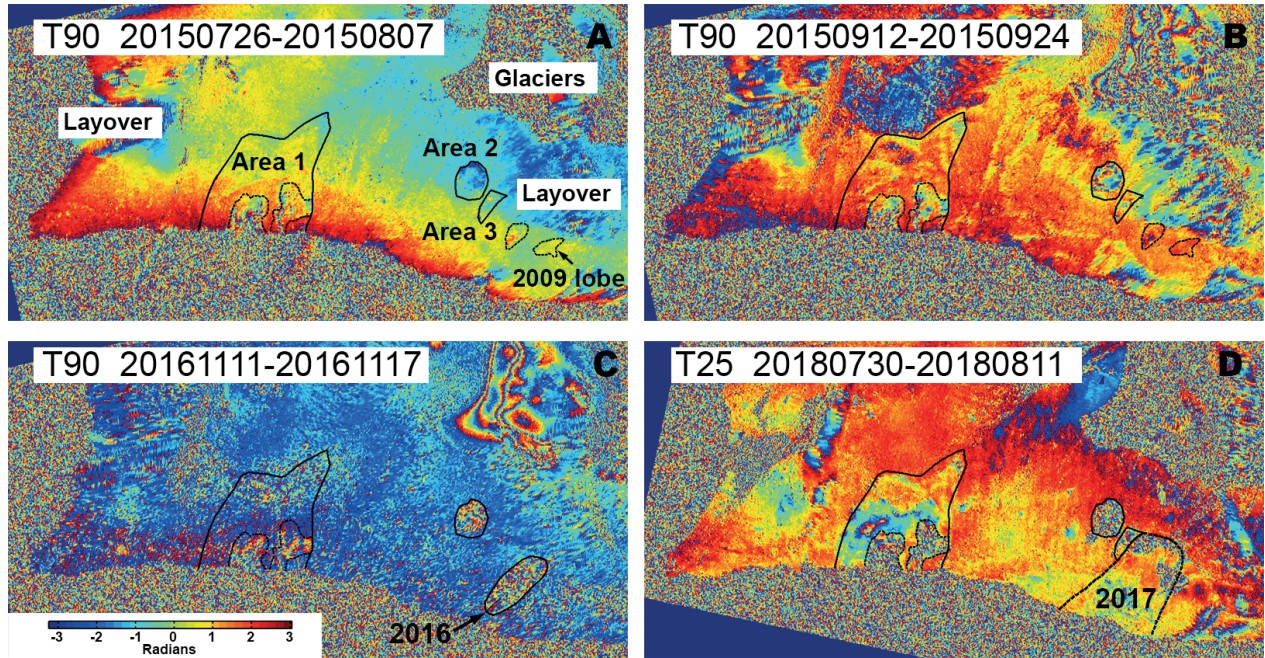

Fig. 5

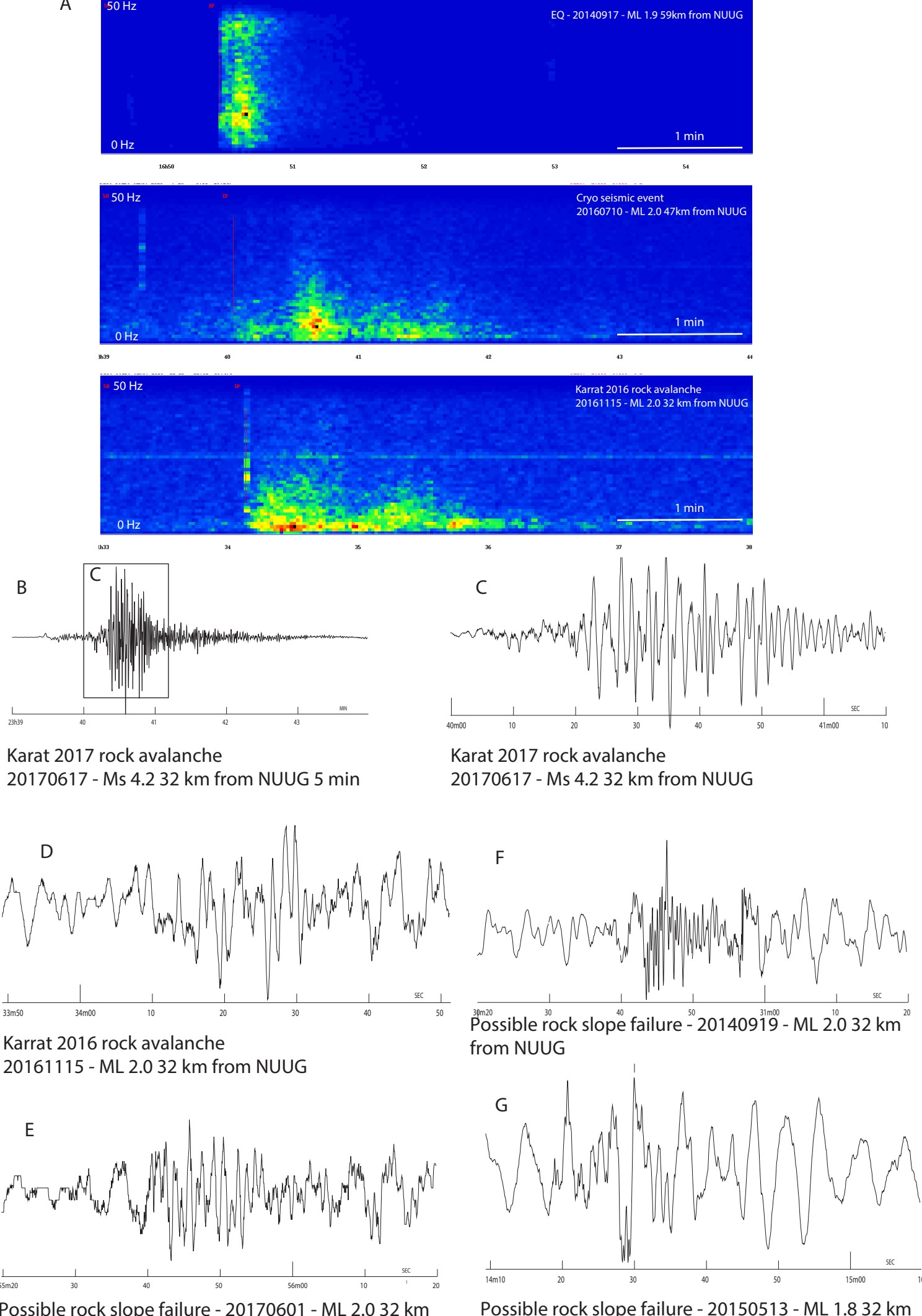

A

50 Hz

0 Hz

EQ - 20140917 - ML 1.9 59km from NUUG

1 min

16h50    51    52    53    54

50 Hz

0 Hz

Cryo seismic event
20160710 - ML 2.0 47km from NUUG

1 min

8h39    40    41    42    43    44

50 Hz

0 Hz

Karrat 2016 rock avalanche
20161115 - ML 2.0 32 km from NUUG

1 min

1h33    34    35    36    37    38

B

C

23h39    40    41    42    43    MIN

Karat 2017 rock avalanche
20170617 - Ms 4.2 32 km from NUUG 5 min

C

40m00    10    20    30    40    50    41m00    10    SEC

Karat 2017 rock avalanche
20170617 - Ms 4.2 32 km from NUUG

D

33m50    34m00    10    20    30    40    50    SEC

Karrat 2016 rock avalanche
20161115 - ML 2.0 32 km from NUUG

F

30m20    30    40    50    31m00    10    20    SEC

Possible rock slope failure - 20140919 - ML 2.0 32 km
from NUUG

E

55m20    30    40    50    56m00    10    20    SEC

Possible rock slope failure - 20170601 - ML 2.0 32 km
from NUUG

G

14m10    20    30    40    50    15m00    10    SEC

Possible rock slope failure - 20150513 - ML 1.8 32 km
from NUUG

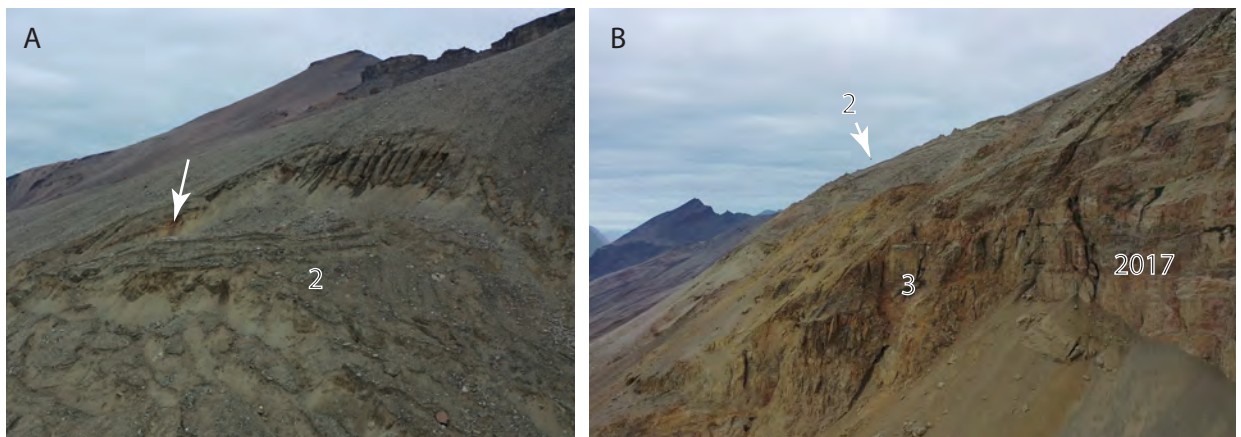

Fig. 7

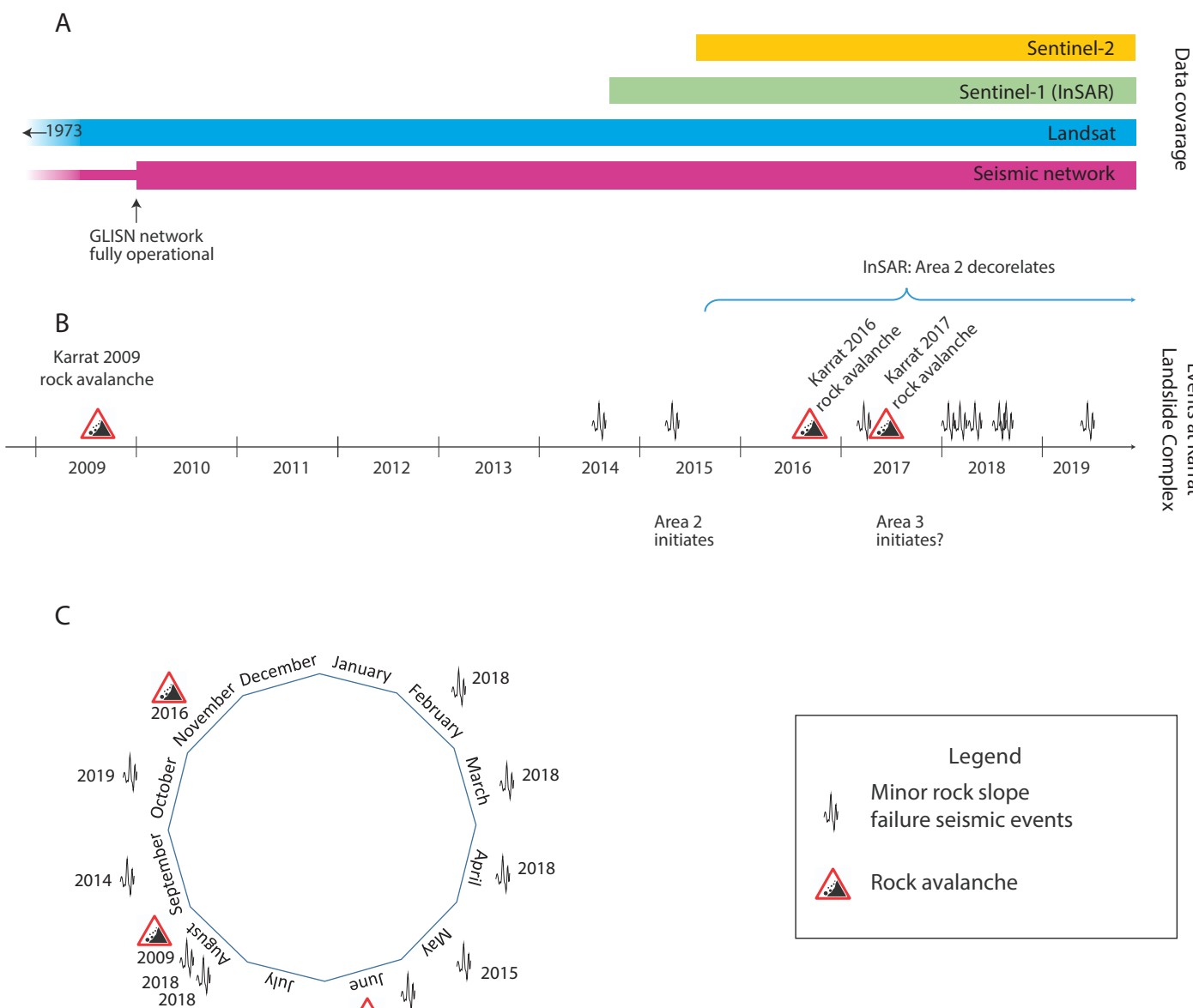

Fig. 8