# Peer review of "Evolution of events before and after the 17 June 2017 rock avalanche at Karrat Fjord, West Greenland – a multidisciplinary approach to detect and locate unstable rock slopes in a remote Arctic area"

_Earth Surface Dynamics, 2020_

## Referee Comment (RC1) · Anonymous Referee #1 · 12 May 2020

I read a multi-approach study on mass wasting activity in West Greenland, revolving around a major landslide that happened in summer 2017. The authors combine a large set of interesting, valuable, and complementary data sources to shed light onto the evolution and properties of an apparently notoriously active hillslope. I found the findings and benefits of combining the different sources of information intriguing in general and see a benefit in the approach itself. However, at the same time I think the study suffers from several, partly severe flaws that challenge a publication in its current shape.

[Figure]

At the current state, I see the general lack of a proper research question that would generate value and impact beyond just the case study scope. Currently, the article generates just incremental new knowledge. The workflow has already been presented in an earlier study. The main 2017 event itself and some of the secondary events have been described in detail by other studies (which have been adequately referenced). So, without describing a new method or workflow and without providing substantial original insight to previously reported events (apart from a few minor and indeed valuable prior and posteriour slope failures), the scope of the study shrinks to the two research questions the authors present at the end of the introduction: i) understand the processes that lead to 2017 event and ii) evaluate the risk of future events in the area. Sadly, in my opinion, these two research questions, despite being relevant and timely, cannot be be addressed with the methods and data the authors have presented. Neither the drivers nor the triggers of the 2017 event could be constrained with sufficient rigour, and also the trends of future activity are a matter of speculation. Investigating drivers and triggers would at least require additional data on for example meteorological, geophyiscal and ground properties of the area, none of which are presented. Constraining future areas at risk would need to go significantly beyond qualitative statements of potential rock texture characteristics or speculations about future permafrost trends. Thus, I recommend the authors develop one strong research question they can address with their data, and the data is indeed very promising, given that the analysis goes beyond the currently very descriptive nature (see below).

The description of methods is in many ways not rigorous enough to allow a solid judgement of whether or not the data can support the stated claims. As a few examples: There is no detail about the software and workflow used to perform the InSAR analysis. There is no information on the seismic data handling and analysis (e.g., signal preprocessing, detection of events, location of events, magnitude estimation, description/analysis of event signals; from the data presented in figure 5, it looks like the raw seismograms were inspected, without deconvolution, without filtering, without description of the spectral properties and their evolution, without inversion of the data for

forces or other target variables). There is only very diffuse information on how optical remote sensing data was interpreted to identify features, no software or workflows are described. Thus, as a reviewer I mainly have to guess what might have been done. And especially for the first two points this is very unsatisfying and far from good practice. Thus, I strongly suggest the authors provide substantial detail on their workflows and software environments, to allow readers (and reviewers) to judge the validity and rigour of their analyses.

Similar to the above point, several results and conclusions appear not supported from the presented methods. Examples are bedding characteristics (constrained from geological map, own mapping, UAV data,...?), triggers of events, permafrost influences and trends, sliding plane angles (how evaluated, what are uncertainties, and so on), volume calculations (how constrained, how processed, what are the uncertainties). This does not mean I do not believe in the results, it just means the authors must give more detail about how the material they present has been generated to avoid such questions as above in the first place.

Overall, the presented material is very descriptive and subjective. The descriptive character is in itself not necessarily a flaw, but it is a pity that the authors did not go beyond that stage of data presentation, as the data would allow for much more quantitative and detailed insights, insights that would massively increase the impact of the study. InSAR data can yield so much more than just colourful pictures (without a legend by the way) and separating areas (of which size and with which degree of overlap to the failed sites?) of decorrelation from areas of coherence (by which measure actually?). Seismic data (see references of what other people have done with seismic data sets) can give so much more insight to the dynamics of mass wasting events (force inversion, volume estimates, duration and evolution,...). In summary, it appears that the authors merely scratch at the surface of their data, although digging a bit deeper would require just a bit more effort (or collaboration with experts in the respective fields). Thus, I recommend more detailed and derivative, quantitative analysis of the valuable data at

hand, an analysis that shall be tightly aligned to the previously developed research question.

The subjective character refers to incomplete presentation of the data. As a Null hypothesis, I would state that there has been mass wasting activity all the time, and not just during the periods presented in table 2. To reject that Null hypothesis, a reader needs to be convinced that the data (mainly optical imagery, seismic waveforms, InSAR-based surface change) indeed shows no activity/change throughout the period covered by the data. What is the coherence like for all InSAR scences? Can mobility hotspots be identified, tracked and quantified through time? What does the seismic data say about hillslope-related activity since about 2000? How many seismic events were detected and located, and how? I strongly encourage the authors to put more effort in turning the qualitative data into appropriate quantitative data, which would allow more robust, rigorous and valuable insight to the dynamics of this very interesting study area.

In summary, I indeed believe there is valuable information and insight hidden in the manuscript. But at the current state this potential value is cluttered and diffuse, and the authors must put significant effort in sculpting the valuable information from the big chunks of raw data they are currently presenting. In addition, the study must develop a clear research question that can be pursued by the data and analyses applied to it. Given that these above points can be addressed in the near future, I would be very happy to give more detailed feedback on the methods, results and implications of the study.

---

## Author Comment (AC1) · 12 May 2020

Dear reviewer Thank you for your insightful and constructive comments that raises many good points. We will await further comments and address these in detail when the discussion closes. Kind regards Kristian
* * *

---

## Referee Comment (RC2) · Anonymous Referee #2 · 8 Jun 2020

General comments: The topic is interesting and important, but the article needs major revisions before it is published. The article needs to be better organized. The intro needs something about what you knew before the landslide and what you didn't and where else is at risk of these kinds of landslides/tsunami. The physiographic setting needs more than a couple of sentences on the bedrock geology. Need to present: climate, past glaciations, i.e, surficial geology, permafrost distribution, bathymetry, and why a tsunami wave would have made it that far. The methods need a work flow diagram showing what you did first and what you did last. Field verification is likely

last. The explanations for each method are too general. You also need to talk about which software packages you used for each method. You should mention the number of images that were used to give you the results you are presenting. Eg DInSAR monitoring...how many images? Also, explain decorrelation, a variation of colour is shown but no one can tell how much change has actually occurred because there is no colour bar indicating the colours. Your figures need to be referred to succinctly in the text. You should not mention Fig 5 until you mention the ones ahead of it. The same for Fig 2e before a,b,c,d. And if you can't do that, reorganize the figs according to your text. You should always cite references in text chronologically. Overall, the article has potential to show what can be done remotely, but it is not ready yet as it is not clear how well you have met the objectives. Specific comments: They are on the marked copy that I have uploaded. One refence is listed in the ref list but not in text. Hovikoski et al. For the figures, I have made comments directly on the caption.

Please also note the supplement to this comment:
https://esurf.copernicus.org/preprints/esurf-2020-32/esurf-2020-32-RC2-supplement.zip

---

## Short Comment (SC1) · 9 Jun 2020

Dear reviewer Thank you for your in depth review and constructive comments. We will await the closing of the discussion and address the issues you have raised in detail then. Kind regards Kristian Svennevig

---

## Short Comment (SC2) · 17 Jun 2020

Dear reviewer Thank you for your review and constructive comments. We will await the closing of the discussion and address the issues you have raised in detail then. Kind regards Kristian Svennevig

---

## Referee Comment (RC4) · Anonymous Referee #4 · 19 Jun 2020

The manuscript describes a cluster of unstable rock slopes and rock avalanches on one slope of Karrat fjord, one of those the June 17th rock avalanche that caused a displacement wave and loss of life in the 30 km distant village of Nuugaatsiaq, W Greenland. The manuscript summarizes available remote sensing data from seismic stations, satellite and air and includes limited field observations. The manuscript summarizes data on published and unpublished events. In the discussion the authors ague that remote sensing data and databases on events are a good tool to predict large rock slope failures that might be more catastrophic. In addition, the authors discuss

conditioning mechanisms and suggest that permafrost changes might be one of the contributing mechanisms. I think the manuscript summarizes a large number of new and previously published data to enhance the understanding of rock slope instabilities on a fjord section in the Artic. In this way are the data presented unique for these climatic settings and contribute significantly to the discussion of how large catastrophic slope failures can develop. I think this manuscript is a very valuable contribution to the understanding of rock slope failure development and methods that can be used. However, I see several short comings that have to be addressed prior to publication that I would classify as "mayor revision".

One mayor concern I have with the use of the landslide terminology. Throughout the manuscript there is a constant jump of terms between landslide, rock avalanche, unstable slopes, active areas, slump, slide etc. I think all phenomena described in the manuscript are in rock and none in soil. Soil slides in form of debris flows, soil creeps and others also exist in this environment. However as they are not part of the manuscript I would suggest to limit terminology to "rock avalanche" and "unstable rock slope" as no detailed descriptions (geological profiles) are given to classify the unstable rock slopes more adequately as "slumps", translational, bi translational or other type of rockslides. Similar is my concern with the term "historic" that is not defined in the manuscript but frequently used and exchanged with "recent". My definition of "historic" would be documented in written documents. Thus, although very recent are two of the rockslides not "historic" but reconstructed. In the following I go with my comments in sequential order two times throw the manuscript. In the first round by marking major issues that have to be addressed and in the second round focussing on small issues that can be changed / or have to be changed to improve the manuscript. Abstract: The abstract is badly structured, the newest relevant information come in the second paragraph. The first paragraph only summarizes information that is already published, or methods used. Introduction: The aims are poorly defined. This manuscript is more than a local case study to evaluate the threat or hazard of rock slope failures at Ummiammakku Mountain or the processes that lead to the Karrat 2017 rock avalanche. It is a

first step to develop one method that can help defining the threat/ hazard of rock slope failures in an unhospitable climate with very difficult access. The study also should contribute to the understanding of conditioning mechanisms including permafrost changes. Methods and data: This section gives a good overview of data used but details on each method are too limited. E.g. seismology. It takes the reader to read to the discussion (line 407 – 418) to understand what was done. The methodology section has to be more precise in order that an independent scientist could reconstruct the same results. This is also necessary for the section on InSAR. How processing of data was done with InSAR keeps unclear. Results: I would suggest reorganizing the text blocks of each rock slide or rock avalanche by describing what is today visible, conclude on the process and than reconstruct the event / slide by remote sensing data. Here a bit more description becomes necessary. Why are the deposits of the "confirmed" events "deposits of rock avalanches" but not catastrophic rockslides? The deposits are visible in figure 1-3 but those figures are only tiny. More descriptive documentation should be added which could be placed in a data repository. A detailed data repository would also be enormously beneficial to document the change in remote sensing data for each event. Figure 6 could be added in the data repository as it does not provide essential information to understand the manuscript. Some morphological features are described for the different events/unstable rock slopes. In general only the back scarp somehow easily visible in figures 1-3. Additional material is required, and landforms described should be marked. Some information on the rockslide is given, however the description by far do not allow defining slumps = rotational slide or other types of rock slope failure. So keep it to "unstable rock slope" or go in depth, produce shamtic sections of the instabilities and classify them correctly. Table 2 is confusing as it is unclear what goes into column 1 and 3. In column 1 is a mixture of "interpreted events" Karat 2009 rock avalanche, Karat 2016 rock avalanche and registered events "all seismic events" the Karrat 2017 rock avalanche. Column 3 summarizes, references, interpretation of some of the seismic events or repeats information given in column 1 with other words. This table has to be reorganized. Line 324-325 this should be mapped and shown

somewhere in the result section. This is not a discussion but results of the mapping. Include in figure 1 and make an own figure for this or add into a supplementary data file. Out of the result section it also does not come clear if the remaining slopes in the fjord were mapped and no landslides were detected or if no sign of large landslides was seen rapidly and thus the slopes not mapped. This information is essential for the discussion. Line 343-346 This is rather a result and not a discussion. A nice figure could be added or this statement should be documented in a supplementary data file. Large part of the description on the seismological signature of a landslide should not go into the discussion but into the result chapter including figure 5. Discussion: Work flow: this work flow is clearly new and it was developed based on the remoteness of the environment. However, it should be discussed against workflows form other environments and what is the improvement of workflow that could also give advantages in other settings. Trigger/conditioning mechanism: Effects of static, dynamic conditioning factors and triggering have long been discussed e.g. Glade and Crozier 2005, Hermanns et al., 2006a and others and the discussion here could follow those classes as structural geology clearly is a conditioning factor here while permafrost changes is very likely one. The study does not contribute to the discussion of triggering and rock fatigue or a form of widening of the instability can be discussed. Look into the wide literature of progressive rock slope failure for references. The discussion on permafrost is relatively poor in respect with recently published papers on the topic. I think that the hypothesis of permafrost degradation is valid however it should be discussed based on other publications, e.g.: McColl, 2012; Ballantyne and Stones, 2013; Böhme et al., 2015; Hilger et al., 2018; Kuhn et al., 2019 The same counts for repeated failure from the same slope. There is a vast literature discussing the relation between repeated failures: Grimstad, 2005; Hermanns et al., 2006b; Willenberg et al., 2008; Hilger et al., 2018 The discussion starts with referring to the work by Krautblatter et al., 2013. This paper summarizes different effects of permafrost change on rock slope stability. The discussion does not include any details on that.

Small comments: Line 52: Risk is hazard x consequences, this manuscript does not

include a consequence analysis rather use threat or hazard if you include volume, impact area and likelihood analysis. Line 131 Figure numbering is out of sequence. Line 195 What is "catastrophic", do you use definition by Hermmans and Longva, 2012 or does your definition include "life loss" Line 210 "candidate" is jargon. Line 214 A "scarp" is a surface feature "back scarp, minor scarp, secondary scarp" but has no volume, better volume of the source area Line 230: This is not visible in figure 3C. Line 272: Define "initiation" do you mean first surface displacement? Or first failure of rock bridges? Line 294: Should be: "Other seismic activity in the Karrat landslide Complex area that can be related to rock slope deformation." Line 319: Erosion is much more and you do not discuss landslide versus other type of erosion here better simple "rock slope failure activity" Line 334: Formulate a bit more cautious "with interpreted seismic characteristics" Line 340: seen in InSAR and seismic events that can be associated to it. Line 344: Use "likelihood" instead of "risk" Line 362: This is like multiple rock avalanches at Loen in Norway in the 20th century. Line 363: "could be at work" is jargon better: contribute Line 376: Magnin et al., 2019 would be an ideal reference here (published in the same journal :-) ) Line 383: You described that structures and their orientations are important, are those known for slopes in neighbouring fjord systems? Line 441: "being alert to smaller events in a known landslide area is crucial" better "might be crucial" there are multiple failures that are not proceeded by small failures but rather by opening of cracks – see literature.

Please also note the supplement to this comment:
https://esurf.copernicus.org/preprints/esurf-2020-32/esurf-2020-32-RC4-supplement.pdf

---

## Short Comment (SC3) · 21 Jun 2020

Dear reviewer Thank you for your detailed review and constructive comments. We will await the closing of the discussion and address the issues you have raised in detail then.

Kind regards Kristian Svennevig

---

## Author Comment (AC2) · 24 Sep 2020

First and foremost, we thank you for taking the time to review our manuscript and contributing with constructive comments that helped improve the manuscript immensely.

We have made our aim/research question clearer in the title, abstract and introduction. "Our aims with this study are twofold: to understand the processes that led to the disastrous Karrat 2017 rock avalanche and the continued threat from the area, and to explore our ability to detect and locate rock slope failures and ultimately to assess

the associated hazard in an unhospitable climate with very difficult access" This also goes towards answering your suggestion to take the various datasets (namely seismic and InSAR) further and use them to their full potential which of course could be very interesting. However, A detailed examination of the 2017 Karrat rock avalanche using these methods are beyond the scope of this paper. We set out to develop a way to examine the slopes in these inaccessible areas of the world, and during that course have described the evolution of the Karrat Landslide Complex and raised some questions to be answered by one or more in depth studies hopefully also including detailed fieldwork and dating of previous rock slope failures. The authors do not have the resources, nor the scientific background (in some areas) to do the detailed work you suggest but would be very open to cooperation on the area as there are surely much more to learn from this spectacular landslide complex.

Svennevig et al 2019 that you refer to is a short preliminary paper on a single event, demonstrating that the landslide area is still active. With the present paper we go beyond this work and show that we can describe the recent evolution of the Karrat Landslide Complex.

For a detailed overview of the changes made we refer to the the reviewer comments overview document attached here and the updated manuscript we resubmit.

Kind regards, on behalf of the authors Kristian Svennevig

Please also note the supplement to this comment:
https://esurf.copernicus.org/preprints/esurf-2020-32/esurf-2020-32-AC2-supplement.pdf

---

## Author Comment (AC3) · 24 Sep 2020

Thank you for taking the time to review our manuscript. We have accommodated your suggestions and it has substantially improved the manuscript.

We have restructured the paper significantly and made the aim of the paper more clear in the title, abstract and introduction: "Our aims with this study are twofold: to understand the processes that led to the disastrous Karrat 2017 rock avalanche and the continued threat from the area, and to explore our ability to detect and locate rock

slope failures and ultimately to assess the associated hazard in an unhospitable climate with very difficult access" We have furthermore updated the physiography according to your suggestions but have not gone into a detailed description of the bathymetry and tsunami as this is beyond the scope of this paper. The tsunami has been described and modelled by Paris et al 2019.

We have elaborated on the methodology describing the workflow in more detail and described the individual components in more detail.

For a detailed overview of the changes made we refer to the reviewer comments overview document attached here and the updated manuscript we resubmit.

Kind regards on behalf of the authors Kristian Svennevig

Please also note the supplement to this comment:
https://esurf.copernicus.org/preprints/esurf-2020-32/esurf-2020-32-AC3-supplement.pdf

---

## Author Response (AR1)

Letter to editor

Dear prof. dr. Michael Krautblatter

Thank you for considering our manuscript for review in Esurf and thank you for your understanding regarding the extension of the resubmission deadline.

We have chosen to follow the suggestions of reviewer 4 closely as we found this review the most thorough, constructive and challanging. We have, in addition, also accommodated most of the changes suggested by the other reviewers. How we respond to the various reviewers main criticism can be seen in the individual answer to reviewers. For a detailed overview of the changes made we refer to the "reviewer comments overview" table in the following and the updated manuscript with changes highlighted at the end of this document.

In response to reviewer 1s comment who raises some "grave" issues we would like to address the following:

"In summary, it appears that the authors merely scratch at the surface of their data, although digging a bit deeper would require just a bit more effort (or collaboration with experts in the respective fields)"

We say that an in-depth examination of all the various datasets was never our intent. We "merely" set out to detect and locate unstable rock slopes and rocks slope failures in this challenging setting: the harsh arctic environment where we really have no alternative and where fieldwork is difficult and expensive. We have addressed this misunderstanding by making the aim of our paper more clear in the title, abstract, introduction and conclusion:

"Our aims with this study are twofold: 1) to understand the processes that led to the disastrous Karrat 2017 rock avalanche and the continued threat from the area, 2) and to explore our ability to detect and locate rock slope failures and ultimately to assess the associated hazard in an unhospitable climate with very difficult access"

Reviewer 1 furthermore suggests that the method have been published already (in Svennevig et al 2019). To this we say that Svennevig et al 2019 is a short preliminary paper on a single event the main focus of which is to demonstrate that the landslide area is still active (which was not published prior to this). With the present paper we go beyond this work and show that we resolve the recent history of the Karrat Landslide Complex in some detail and identify a number of active unstable rock slopes.

We look forward to hearing from you again

Kind regards, on behalf of the authors

Kristian Svennevig

**Overview of reviewer comments**

In the following table the comments from the reviewers have been listed as they appear in the text along with out answer to them. We further refer to the manuscript with tracked changes.

| | Reviewer | Reviewer comment | Author answer |
|---|---|---|---|
| | | **General comments** | |
| | R1 | I recommend the authors develop one strong research question they can address with their data, and the data is indeed very promising, given that the analysis goes beyond the currently very descriptive nature | We have made clear what the aim of the paper |
| | R2 | Detailed comments given in "esrf-2020-32 R2_comments" | We have accommodated the numerous minor corrections and suggestions named in the document |
| | R2 | Your figures need to be referred to succinctly in the text. You should not mention Fig 5 until you mention the ones ahead of it. The same for Fig 2e before a,b,c,d. | We have edited the figs (added new fig 2) and corrected the figure referencing |
| | R2 | You should always cite references in text chronologically. | We have corrected this throughout the paper |
| | R2 | Overall, the article has potential to show what can be done remotely, but it is not ready yet as it is not clear how well you have met the objectives. | We have made the aim of the paper more clear (to detect and locate unstable rock slopes and rock slope failures) |
| | R4 | Mayor concern I have with the use of the landslide terminology | We have limited the terminology to "rock avalanche" and "unstable rock slope" as suggested, and exchanged the term "landslide" with "rock slope failure" |
| | R3, R4 | the terms "historic" and "prehistoric" confusing. | We now describe the activity as "recent" or "older" |
| | R4 | 17 specific minor comments (see RC4) | We have accommodated all the changes |
| | | | |
| | | **Abstract:** | |
| | R4 | The abstract is badly structured. | We've restructured the abstract |
| | | | |
| | | **Introduction** | |
| | R2 | The intro needs something about what you knew before the landslide and what you didn't and where else is at risk of these kinds of landslides/tsunami | We have elaborated on the state of knowledge before the landslide, this is also done in the start of the results section |
| | R2 | The physiographic setting needs more than a couple of sentences on the bedrock geology. Need to present: | We have expanded the physiographic setting. We do not address the tsunami wave. This is done by Paris *et al*. (2019) |

| | | | |
|---|---|---|---|
| | | climate, past glaciations, i.e, surficial geology, permafrost distribution, bathymetry, and why a tsunami wave would have made it that far | |
| | R4 | The aims are poorly defined: [he suggests] It is a first step to develop one method that can help defining the threat/ hazard of rock slope failures in an unhospitable climate with very difficult access. The study also should contribute to the understanding of conditioning mechanisms including permafrost changes. | We have made clear what the aim of the paper in the abstract and introduction. We have elaborated on the conditioning mechanisms in the discussion but a detailed discussion is beyond the scope of this study as we do not contribute with dating on old rock slope faliures and with data on the permafrost conditions of the slope |
| | | | |
| | | **Methods and data** | |
| | R1 | There is no detail about the software and workflow used to perform the InSAR analysis. | More details on the InSAR processing have been added. |
| | R1 | There is no information on the seismic data handling and analysis (e.g., signal preprocessing, detection of events, location of events, magnitude estimation, description/analysis of event signals; from the data presented in figure 5, it looks like the raw seismograms were inspected, without deconvolution, without filtering, without description of the spectral properties and their evolution, without inversion of the data for forces or other target variables) | We have added this to the method section and described why we use this approach. |
| | R1 | There is only very diffuse information on how optical remote sensing data was interpreted to identify features, no software or workflows are described. | We have elaborated on the (very simple) workflow used for the optical images |
| | R1 | InSAR data can yield so much more than just colourful pictures (without a legend by the way) and separating areas (of which size and with which degree of overlap to the failed sites?) of decorrelation from areas of coherence (by which measure actually?) | Detailed InSAR analysis is beyond the scope of this paper. The methodology has been described more clearly. We use InSAR data to identify individual events, not to map long-term deformation rates, thus the two-pair interferograms are the main data for our analysis. More detailed analysis, including multi-temporal InSAR will be presented in a separate paper. |
| | R1 | Seismic data (see references of what other people have done with seismic data sets) | Detailed analysis of the seismic signal is beyond the scope of this paper. We use the |

| | | | |
|---|---|---|---|
| | | can give so much more insight to the dynamics of mass wasting events (force inversion, volume estimates, duration and evolution,...) | seismic data to detect events, namely the exact timing. |
| | R2 | The methods need a workflow diagram showing what you did first and what you did last | The methodology and workflow have been described more clearly. Composing a workflow diagram is not straight forward as there are multiple ways into the workflow. |
| | R2 | The explanations for each method are too general | We have elaborated on the method descriptions |
| | R2 | Eg DInSAR monitoring… how many images? Also, explain decorrelation, a variation of colour is shown but no one can tell how much change has actually occurred because there is no colour bar indicating the colours. | The methodology has been described more clearly, including information about the number of images processed. See also reply to R1 above. |
| | R3 | It is stated that earthquake location uncertainties are up to 50 km, but what are typical "average" uncertainties? And what are typical magnitudes of the recorded events? | The typical uncertainties are dependent on the number of stations recording the event (tied with the magnitude), and with the station spacing. They are from under 10 km for large events to up to 50 km for smaller events. We record events in Greenland from under ML 1 to over ML 6.0 The events discussed in this paper (bar the mail 2017 rock avalanche event) are typically ML 1.2 – 2.7 – se table 2 in the ms. |
| | R4 | details on each method are too limited. The methodology section has to be more precise in order that an independent scientist could reconstruct the same results | The methodology has been described more clearly. |
| | R4 | seismology. It takes the reader to read to the discussion (line 407 – 418) to understand what was done | The seismology section have been restructured |
| | R4 | How processing of data was done with InSAR keeps unclear. | The methodology has been described more clearly. |
| | | | |
| | | **Results** | |
| | R1 | are bedding characteristics (constrained from geological map, own mapping, UAV data,...?), | Bedding trends are from field measurements. We have clarified this in the results |
| | R1 | Volume calculations (how constrained, how processed, what are the uncertainties) | The volumes are constrained by two DEMs subtracted from each other as (now more clearly) stated in the text. |

| | | | |
|---|---|---|---|
| | R1 | sliding plane angles (how evaluated, what are uncertainties, and so on) | The sliding plane is covered by deposits but we infer it from the dip of the bedding in the area. We have clarified this in the text |
| | R3 | Page 7: The 2009 and 2016 rock avalanches have similar volume, but quite different magnitudes (2.7 vs. 2.1). I would be interested in the authors' view on what may be the reason for this discrepancy | Both magnitudes are consistent within the stations recording the events. The plot (end of this document) of the spectra ( 8 min time and frequencies from 0-10Hz in both cases) of the vertical component from SUMG station show that the 2009 ML 2.7 (top) happened in a short time period and concentrated, while the 2016 ML 2.0 (bottom) was much more diffuse and took longer. So even though the two events happened at the same location and the volume and scar look similar, the timing within the events was quite different. A short concentrated fall gives rise to a higher magnitude in the seismic signature than a more diffuse events over longer time. Ideally, this should not be the case as magnitude aims at representing the energy released in the event. Here we use the local magnitude (ML) which is designed for tectonic earthquakes and thus does not represent the full energy release of a non-tectonic event. Very interesting observation from reviewer 3 - thank you. |
| | R4 | I would suggest reorganizing the text blocks of each rock slide or rock avalanche by describing what is today visible, conclude on the process and then reconstruct the event / slide by remote sensing data. Here a bit more description becomes necessary. | We have made an introductory paragraph to the results section to describe more clearly what have been done |
| | R4 | More descriptive documentation should be added which could be placed in a data repository. A detailed data repository would also be enormously beneficial to document the change in remote sensing data for each event. Figure 6 could be added in the data repository as it does not provide essential information to understand the manuscript | We did not find it necessary to compile the data in a repository as we clearly describe what data has been used to identify individual events and as all of the data is freely available through the sources listed in the method and data availability sections. |
| | R4 | Some morphological features are described for the different events/unstable rock slopes. In general only the back scarp somehow easily visible in figures 1-3. | We have added a new fig 2 and updated the other figures to address this |

| | | Additional material is required, and landforms described should be marked. | |
|---|---|---|---|
| R4 | | Some information on the rockslide is given, however the description by far do not allow defining slumps = rotational slide or other types of rock slope failure. So keep it to "unstable rock slope" or go in depth, produce shamtic sections of the instabilities and classify them correctly. | We have limited the terminology to "rock avalanche" and "unstable rock slope" as suggested. |
| R4 | | Table 2 is confusing as it is unclear what goes into column 1 and 3. In column 1 is a mixture of "interpreted events" Karat 2009 rock avalanche, Karat 2016 rock avalanche and registered events "all seismic events" the Karrat 2017 rock avalanche. Column 3 summarizes, references, interpretation of some of the seismic events or repeats information given in column 1 with other words. This table has to be reorganized. | We have reorganized the table. |
| | | | |
| | | **Discussion** | |
| R4 | | Line 324-325 this should be mapped and shown somewhere in the result section. This is not a discussion but results of the mapping. Include in figure 1 and make an own figure for this or add into a supplementary data file. Out of the result section it also does not come clear if the remaining slopes in the fjord were mapped and no landslides were detected or if no sign of large landslides was seen rapidly and thus the slopes not mapped. This information is essential for the discussion | We have made a paragraph on "Field observations and sign of previous activity" in the start of the results section to accommodate this. We have added sentence about screening of the surrounding the KLC. We have added e new fig 2 showing examples of previous activity |
| R4 | | Large part of the description on the seismological signature of a landslide should not go into the discussion but into the result chapter including figure 5. | We have rewritten this section and moved it to the method and result section |
| R4 | | Line 343-346 This is rather a result and not a discussion. A nice figure could be added or this statement should be documented in a supplementary data file. | We have removed this statement form the discussion |
| R3 | | I agree on the last sentence ": : : It is an effective tool for identifying and investigating active landslide areas, but actual field validation is necessary in order to further assess the risk", but it needs elaboration. What can we | We have elaborated that field visits are necessary, especially to constrain the structural setting of the slope. |

| | | | |
|---|---|---|---|
| | | obtain from field data, that we cannot see remotely? And how does that contribute to risk assessment? (and should it actually rather be hazard assessment?) | |
| | R3 | Page 13, first paragraph: have you compared spectral plots of cryogenic seismic events and small landslide events? Could such plots be added to Figure 5? | Figure 5 (now fig 5) has been revised, and spectral plots added. |
| | R4 | Work flow: this work flow is clearly new and it was developed based on the remoteness of the environment. However, it should be discussed against workflows form other environments and what is the improvement of workflow that could also give advantages in other settings. | We have added a discussion and references to accommodate this |
| | R4 | Trigger/conditioning mechanism: Effects of static, dynamic conditioning factors and triggering have long been discussed e.g. Glade and Crozier 2005, Hermanns et al., 2006a and others and the discussion here could follow those classes as structural geology clearly is a conditioning factor here while permafrost changes is very likely one. The study does not contribute to the discussion of triggering and rock fatigue or a form of widening of the instability can be discussed. Look into the wide literature of progressive rock slope failure for references. | Based on our limited (mainly remotely sensed) data a detailed discussion on the trigger/conditioning mechanisms and contribution of permafrost degradation is beyond the scope of this paper. We have rewritten and expanded this part of the discussion with some of the suggested references but avoided going into a detailed discussion |
| | R4 | The discussion on permafrost is relatively poor in respect with recently published papers on the topic. I think that the hypothesis of permafrost degradation is valid however it should be discussed based on other publications, e.g.: McColl, 2012; Ballantyne and Stones, 2013; Böhme et al., 2015; Hilger et al., 2018; Kuhn et al., 2019 | See above |
| | R4 | The same counts for repeated failure from the same slope. There is a vast literature discussing the relation between repeated failures: Grimstad, 2005; Hermanns et al., 2006b; Willenberg et al., 2008; Hilger et al., 2018 | See above |
| | R4 | The discussion starts with referring to the work by Krautblatter et al., 2013. This paper summarizes different effects of permafrost change on rock slope stability. The | See above |

| | | discussion does not include any details on that. | |
| --- | --- | --- | --- |
| | | **Conclusion and outlook** | |
| | R3 | Page 14, 2nd paragraph: I do not agree that being alert to smaller landslide events will mitigate the risk of large, tsunamigenic events, though it may allow for evacuation of exposed populations before a large event. Consider rephrasing. | We have rephrased the conclusion |

Plot of spectra:

[revised manuscript text omitted]

The DInSAR analysis can be performed either using two-pass interferograms, i.e. based on the difference in interferometric phase between pairs of acquisitions, or using time-series analyses of the of available acquisitions, i.e. based on the phase difference of pixels with persistent electromagnetic properties throughout the analysed time span. The deformation at the Karrat Landslide Complex is highly non-linear due to the rock avalanches in 2009, 2016 and 2017, and deformation rates are high causing decorrelation in some areas. Therefore, we choose to use two-pass interferograms for this study, usually with 6 or 12 days between acquisitions, to aid the investigation of the timing of the different events in the area. ADD SENTENCE ABOUT NUMBER OF IMAGES PROCESSED. The interferograms were formed using the SARPROZ software (Perissin et al., 2007). or this study both 6 and 12 days differential interferograms were used. The topographic contribution to the interferometric phase was removed using ArcticDEM version 2.0 (Porter et al., 2018). For interferograms following the June 2017 landsliderock avalanche, the ArcticDEM was locally corrected with a DEM

(digital elevation Model) derived from oblique photogrammetry collected in the summer of 2017 (courtesy of E.V. Sørensen, GEUS).

Two Sentinel-1 satellite tracks cover the Karrat area during the period of interest: ascending track 90 (available from October 2014 and onward) and descending track 25 (available from July 2017 and onward). The viewing geometry of track 25 is bestwell suited for detecting movements on the slopes in our regionarea of interest. Unfortunately, large parts of the steep slopes that failed in 2016 and 2017 dip steeply toward the radar in cannot be observed with the viewing geometry of ascending track 90, causing problems with so-called foreshortening and layover. Thus, deformation that may have led up to the main event in 2017 cannot be observed with the available SAR data from that covers the pre-failure time period.

[revised manuscript text omitted]

* * *
**formaterede:** Ikke Fremhævning

**Feltkode ændret**

**formaterede:** Ikke Fremhævning

**formaterede:** Ikke Fremhævning

**Kommenterede [KS7]:** R4: Effects of static, dynamic conditioning factors and triggering have long been discussed e.g. Glade and Crozier 2005, Hermanns et al., 2006a and others and the discussion here could follow those classes as structural geology clearly is a conditioning factor here while permafrost changes is very likely one. The study does not contribute to the discussion of triggering and rock fatigue or a form of widening of the instability can be discussed. Look into the wide literature of progressive rock slope failure for references.

**Feltkode ændret**

**formaterede:** Ikke Fremhævning

**formaterede:** Ikke Fremhævning

**Feltkode ændret**

**Feltkode ændret**

**Feltkode ændret**

**Feltkode ændret**

**Feltkode ændret**

**formaterede:** Engelsk (USA)

[revised manuscript text omitted]
 unstable slopearea is crucial for the mitigationassessing the hazard of the risk of large, tsunamigenic landslidesrock slope failures. Additionally, by establishing the seismic, InSAR, and optical signatures of precursors for landslidesrock avalanches, it is possible to be alerted of new possible events, both in the Karrat area and elsewhere in isolated arctic areas.~~

~~Although our multi-disciplinary setup has proved to be successful in describing the evolution of the Karrat Landslide Complex, the present study also provides insight to areas of future development: The upgrade of the seismological network in West Greenland in summer 2019 will improve the accuracy of the location of recorded events and thus relatively cheaply improve our understanding of landslide the activity events in the area by helping distinguishing between~~

**Kommenterede [KS12]:** R3: do not agree that being alert to smaller landslide events will mitigate the risk of large, tsunamigenic events, though it may allow for evacuation of exposed populations before a large event. Consider rephrasing

[revised manuscript text omitted]